# The helper NLR immune protein NRC3 mediates the hypersensitive cell death caused by the cell-surface receptor Cf-4

**Jiorgos Kourelis**, **Mauricio P. Contreras**, **Adeline Harant**, **Hsuan Pai**, **Daniel Lüdke**, **Hiroaki Adachi**[¤a], **Lida Derevnina**[¤b], **Chih-Hang Wu**[¤c]*, **Sophien Kamoun**[*]

The Sainsbury Laboratory, University of East Anglia, Norwich Research Park, Norwich, United Kingdom

¤a Current address: Laboratory of Crop Evolution, Graduate School of Agriculture, Kyoto University, Mozume, Muko, Kyoto, Japan
¤b Current address: Crop Science Centre, Department of Plant Sciences, University of Cambridge, Lawrence Weaver Road, Cambridge, United Kingdom
¤c Current address: Institute of Plant and Microbial Biology, Academia Sinica, Nankang, Taipei, Taiwan
* wuchh@gate.sinica.edu.tw (CHW); sophien.kamoun@tsl.ac.uk (SK)

**Data Availability Statement:** All data is included in the manuscript and in supporting information.

**Funding:** This work has been supported by the Gatsby Charitable Foundation (MPC, HP, DL, SK), Biotechnology and Biological Sciences Research

## Abstract

Cell surface pattern recognition receptors (PRRs) activate immune responses that can include the hypersensitive cell death. However, the pathways that link PRRs to the cell death response are poorly understood. Here, we show that the cell surface receptor-like protein Cf-4 requires the intracellular nucleotide-binding domain leucine-rich repeat containing receptor (NLR) NRC3 to trigger a confluent cell death response upon detection of the fungal effector Avr4 in leaves of *Nicotiana benthamiana*. This NRC3 activity requires an intact N-terminal MADA motif, a conserved signature of coiled-coil (CC)-type plant NLRs that is required for resistosome-mediated immune responses. A chimeric protein with the N-terminal α1 helix of Arabidopsis ZAR1 swapped into NRC3 retains the capacity to mediate Cf-4 hypersensitive cell death. Pathogen effectors acting as suppressors of NRC3 can suppress Cf-4-triggered hypersensitive cell-death. Our findings link the NLR resistosome model to the hypersensitive cell death caused by a cell surface PRR.

## Author summary

Just like humans, plants get sick. They get infected by parasites as diverse as fungi, bacteria, viruses, nematodes and insects. And just like humans, plants have an immune system that helps them defend against these parasites. Their first line of defence are disease resistance genes. When a resistance gene in the plant matches a gene in the parasite, the plant immune system is activated and the parasite invasion is restricted. Here, we show that one type of disease resistance gene that functions as a sensor on the cell surface requires other components inside the cell to execute the immune response. Pathogens evolved to suppress this type of immune networks by targeting the hub that connects different types of immune pathways.

Council (BBSRC) (BB/P012574 (Plant Health ISP)) (SK), European Research Council (ERC) (Retooling plant immunity for resistance to blast fungi, grant number 743165) (AH, SK), Japan Society for the Promotion of Plant Science (JSPS) Overseas Research Fellowships (grant number 1040031) (HA), Deutsche Forschungsgemeinschaft (DFG) Walter Benjamin Program (project number 464864389) (DL), and BASF Plant Science (BASF Plant Science) (JK, SK). The funders had no role in study design, data collection and analysis, decision to publish, or preparation of the manuscript.

**Competing interests:** I have read the journal's policy and the authors of this manuscript have the following competing interests: JK and SK receive funding from industry and have filed patents on NLR biology.

# Introduction

Recognition of pathogens by the plant innate immune system predominantly relies on cell surface immune receptors and intracellular nucleotide-binding and oligomerization domain (NOD)-like leucine-rich repeat receptors (NLRs). The activation of these receptors upon recognition of their cognate ligands triggers specific responses which ultimately result in activation of downstream immune pathways and disease resistance. The ligand sensing mechanisms are either by direct binding of a pathogen-derived or pathogen-induced ligand to the receptor, or by indirect recognition requiring additional host proteins [1]. Once cell-surface receptors and intracellular NLRs are activated, both classes of receptors induce common outputs, such as $Ca^{2+}$ influx, reactive oxygen species (ROS) production and mitogen activated protein kinase (MAPK) activation. However, the molecular mechanisms by which receptor activation results in common downstream immune signaling are not as well understood. The extent to which the downstream signaling of cell surface immune receptors and intracellular NLRs differs and converges is one of the main open questions in the field of plant immunity [2].

The plant NLR family is defined by a central NB-ARC (nucleotide-binding domain shared with APAF-1, various R-proteins and CED-4) domain and at least one other domain [3]. Plant NLRs can be subdivided in four main subclades characterized by distinct N-terminal domains. TIR-NLRs contain a catalytically active N-terminal Toll/interleukin-1 receptor (TIR) domain, while $CC_R$-NLRs, $CC_{G10}$-NLRs, and CC-NLRs contain distinct RPW8-type, G10-type, or Rx-type coiled coil (CC) domains, respectively. Several breakthroughs have been made in recent years in understanding the molecular mechanisms of recognition and signaling of plant NLRs [4]. NLR activation upon ligand recognition can result in an exchange of adenosine diphosphate (ADP) to adenosine triphosphate (ATP) in the NB-ARC domain and the formation of an oligomeric complex, termed a "resistosome". The Arabidopsis CC-NLR HOPZ-ACTIVATED RESISTANCE 1 (ZAR1) forms a pentameric resistosome structure in which the N-terminal α1 helix extends out to form a funnel which acts as a $Ca^{2+}$-permeable channel [5–7]. Likewise, upon activation the wheat CC-NLR Sr35 also forms a pentameric resistosome structure which acts as a $Ca^{2+}$-permeable channel, highlighting a common mechanism by which CC-NLRs function [8]. Like CC-NLRs, the $CC_R$-type helper-NLRs N REQUIREMENT GENE 1.1 (NRG1.1) and ACTIVATED DISEASE RESISTANCE 1 (ADR1) also generate $Ca^{2+}$-permeable channels upon activation [9]. The TIR-NLRs RECOGNITION OF PERONOSPORA PARASITICA 1 (RPP1) and RECOGNITION OF XopQ1 (Roq1) form tetrameric resistosome complexes upon activation [10,11]. The oligomerization of the TIR domain results in nicotinamide adenine dinucleotide nucleosidase (NADase) activity and the release of a plant-specific variant cyclic ADP-ribose (v-cADPR) [12,13]. In addition, some plant TIR domains can form a distinct oligomeric form with 2′,3′-cyclic adenosine monophosphate (cAMP)/cyclic guanosine monophosphate (cGMP) synthetase activity by hydrolyzing dsRNA/dsDNA and this activity is also required for immune signaling [14]. A subset of NLRs, known as functional singletons, combine pathogen detection and immune signaling activities into a single protein [15]. In contrast, many NLRs have functionally specialized over evolutionary time in "sensor" or "helper" NLRs dedicated to pathogen recognition or immune signaling, respectively [15]. Sensor and helper NLRs function together and can form receptor networks with a complex architecture [16,17].

Cell surface receptors are typically divided into two categories: the receptor-like kinases (RKs, also known as RLKs) and the receptor-like proteins (RPs, also known as RLPs). RKs contain a C-terminal intracellular kinase domain for downstream signaling, while RPs contain a small cytoplasmic tail and lack a C-terminal kinase domain. Both RKs and RPs have a single-pass transmembrane domain and a variety of extracellular ligand-binding domains [18]. Upon

ligand perception, RKs typically hetero-oligomerize with other RKs which act as co-receptors. For example, binding of chitin by the Arabidopsis LysM-RK LYSM-CONTAINING RECEPTOR-LIKE KINASE 5 (LYK5) induces hetero-oligomerization with the LysM-RK co-receptor CHITIN ELICITOR RECEPTOR KINASE 1 (CERK1) [19]. Furthermore, recognition of the flg22 peptide of bacterial flagellin by the leucine-rich repeat (LRR)-RK FLAGELLIN-SENSI-TIVE 2 (FLS2) induces hetero-oligomerization with the LRR-RK co-receptor SOMATIC EMBRYOGENESIS RECEPTOR-LIKE KINASE 3 (SERK3, also known as BRASSINOSTER-OID INSENSITIVE 1-ASSOCIATED RECEPTOR KINASE (BAK1)) [20]. Similarly, most LRR-RPs constitutively interact with the LRR-RK SUPPRESSOR OF BAK1-INTERACTING RECEPTOR-LIKE KINASE 1 1 (SOBIR1), and activation upon ligand binding induces the hetero-oligomerization with the LRR-RLK co-receptor SERK3/BAK1 [21–27]. These activated complexes in turn phosphorylate specific downstream receptor-like cytoplasmic kinases (RLCKs) which ultimately relay the immune response [2,28].

While the signaling cascades downstream of cell surface immune receptors and NLRs were thought to be distinct [29], it is now becoming clear that in fact these signaling modules intertwine at different points [30–32]. Cell surface immune signaling was recently shown to be required for the hypersensitive cell death immune response triggered by a variety of NLRs [30,31]. Specifically, in the absence of cell surface immune signaling, the activation of the TIR-NLR pair RESISTANT TO RALSTONIA SOLANACEARUM 1 (RRS1)/RESISTANT TO P. SYRINGAE 4 (RPS4) by the effector AvrRps4 does not result in hypersensitive cell death [30]. Moreover, activation of the $CC_{G10}$-NLRs RESISTANT TO P. SYRINGAE 2 (RPS2) and RESISTANT TO P. SYRINGAE 5 (RPS5) by AvrRpt2 [30,31] and AvrPphB [30], respectively, or the CC-NLR RESISTANCE TO P. SYRINGAE PV MACULICOLA 1 (RPM1) by AvrRpm1 [30] results in diminished hypersensitive cell death in the absence of cell surface immune signaling. The emerging model is that cell surface immune signaling and intracellular NLR signaling mutually potentiate each other by enhancing transcription of the core signaling components of these pathways [30,31]. However, the mechanistic links between cell surface immune receptors and NLRs are not yet known [33]. Additionally, there is a tendency in the field to view cell surface and NLR receptors through a unified lens despite their huge structural and functional diversity [3,18]. The degree to which distinct classes of cell surface and intracellular receptors engage in signaling networks remains unclear.

In addition to their potentiation of cell surface immune signaling, NLRs can also be genetically involved downstream of cell surface immune receptor activation. The lipase-like protein ENHANCED DISEASE SUSCEPTIBILITY 1 (EDS1) forms mutually exclusive receptor complexes with either the related protein PHYTOALEXIN DEFICIENT 4 (PAD4) or SENESCENCE-ASSOCIATED GENE 101 (SAG101) to perceive small molecules produced by activated TIR-NLRs [34,35]. The ligand-bound EDS1/SAG101 and EDS1/PAD4 complexes in turn activate the $CC_R$-type helper-NLRs NRG1 and ADR1 to initiate immune signaling downstream of TIR-NLRs [34–36]. In Arabidopsis, the EDS1/PAD4/ADR1 and, to a lesser extent, EDS1/SAG101/NRG1 modules are also genetically required for a subset of the immune responses triggered by LRR-RPs [32,37] and LRR-RKs [37]. However, a subset of LRR-RPs can trigger hypersensitive cell death upon activation, and this hypersensitive cell death is not affected in mutants of the EDS1/PAD4/ADR1 and EDS1/SAG101/NRG1 pathway [32]. Indeed, the extent to which the subset of LRR-RPs that trigger hypersensitive cell death require NLRs warrants further investigation. Previously, the NLR protein NRC1 (NLR protein required for hypersensitive-response (HR)-associated cell death 1) was implicated in the cell death mediated by the LRR-RLPs Cf-4 [38], LeEIX2 [39], and Ve1 [40]. However, these studies were based on virus-induced gene silencing and, therefore, off-target effects on other NLR genes cannot be ruled out. Indeed, the silencing experiments performed in *Nicotiana benthamiana* were based on a

fragment sequence corresponding to the LRR domain of tomato NRC1 and predated the sequencing of the *N. benthamiana* genome [41]. The sequencing of the *N. benthamiana* genome revealed that *N. benthamiana* does not encode a functional ortholog of the tomato NRC1 gene [41–44]. Instead, the *N. benthamiana* genome contains homologs of NRC1, namely NRC2, NRC3, and NRC4 [16]. Therefore, the precise contribution of NLRs to the hypersensitive response mediated by Cf-4 type receptors has remained unclear, in part due to incomplete silencing of the various *N. benthamiana* NRCs using previously reported constructs [41,45].

The NRC superclade is a subclade of CC-NLRs that has massively expanded in some asterid plants, including the Solanaceae [16]. The NRC superclade consists of sensor NLRs encoded by *R* genes which confer resistance to diverse pathogens, and the NRC (NLR required for cell death) helper CC-NLRs, which mediate immune signaling and the hypersensitive cell death response downstream of effector recognition by the sensor NLRs. NRCs form redundant nodes in complex receptor networks, with different NRCs exhibiting different specificities for their sensor NLRs. Much like ZAR1, the N-terminal α1 helix of NRCs is defined by a molecular signature called the "MADA motif" [46]. NRCs and ZAR1 are therefore classified as MADA-CC-NLRs, a class that includes ~20% of all CC-NLRs in angiosperms [3,46]. In solanaceous species, the NRC network comprises up to half of the NLRome and can be counteracted by pathogen effectors. Recently, Derevnina *et al.*, [47] showed that effector proteins from the cyst nematode *Globodera rostochiensis* and the oomycete *Phytophthora infestans* specifically suppress the activity of NRC2 and NRC3, independently of their sensor NLR partners.

In this study, we took advantage of various combinations of CRISPR/Cas9-induced loss-of-function mutations in the *N. benthamiana* NRCs to revisit the connection between NRCs and cell surface receptors [46,48,49]. We hypothesized that an activated NRC resistosome is required for downstream signaling upon LRR-RP activation in order to produce hypersensitive cell death. To challenge this hypothesis, we used the LRR-RP Cf4 [50], which upon recognition of the fungal pathogen *Cladosporium fulvum* (Syn. *Passalora fulva*) apoplastic effector Avr4 triggers hypersensitive cell death [51]. Using a combination of reverse genetics and complementation, we determined that Cf-4/Avr4-triggered cell death is largely mediated by the *N. benthamiana* NRC3 homolog, thereby clearly defining the contributions of *N. benthamiana* NRC homologs to pathways downstream of the Cf-4 LRR-RP. By using defined mutations in the MADA motif and generating swaps with the Arabidopsis ZAR1 α1 helix, we show that NRC resistosome function is potentially downstream of Cf-4 activation by Avr4. Finally, we show that divergent pathogen-derived suppressors of NRC activity suppress Cf-4/Avr4-triggered cell death. We conclude that Cf-4/Avr4-triggered hypersensitive cell death is mediated by the MADA-CC-NLR NRC3. Divergent pathogens target this node, thereby suppressing both intracellular and cell surface triggered immune responses. NRC3 is conserved in the Solanaceae and likely mediates hypersensitive cell death triggered by a variety of leucine-rich repeat receptor-like proteins. We propose that NRC3 is a core node that connects cell surface and intracellular immune networks.

## Results

### Cf-4/Avr4-triggered hypersensitive cell death is compromised in the *N. benthamiana nrc2/3* CRISPR mutant line

We previously generated multiple mutant lines of *N. benthamiana* that carry CRISPR/Cas9-induced premature termination mutations in various combinations of the *NRC2* (*NRC2a* and *NRC2b*), *NRC3* and *NRC4* (*NRC4a*, *NRC4b*, and *NRC4c*) genes [46,48,49]. These lines (*nrc2/3*, *nrc4*, and *nrc2/3/4*) enabled us to revisit the contribution of NRC helper NLRs to the

hypersensitive cell death caused by the receptor-like protein Cf-4. To do this, we quantified the cell death response following transient expression of Cf-4/Avr4 by agroinfiltration, or as controls the receptor/effector pairs Pto/AvrPto (mediated by Prf; NRC2/3-dependent), Rpi-blb2/AVRblb2 (NRC4-dependent), or R3a/Avr3a (NRC-independent) (**Figs 1** and **S1**) [16]. Cf-4/Avr4-triggered cell-death was significantly reduced in both the *nrc2/3*, and *nrc2/3/4* CRISPR mutant lines as compared to wild-type (**Fig 1A**). By contrast, even though the lipase-like protein EDS1 and the $CC_R$-type helper-NLRs NRG1 and ADR1 are implicated in mediating some responses downstream of cell-surface receptors in Arabidopsis [32,37], Cf-4/Avr4-triggered cell-death was not affected in *N. benthamiana eds1* CRISPR lines (**S2 Fig**) [52]. Cf-4 protein accumulation was not affected in the *nrc2/3/4* CRISPR lines (**Fig 1B**), indicating that the reduction in cell death is not due to differences in Cf-4 protein accumulation between these lines. While Pto/AvrPto-triggered cell death is completely abolished in the *nrc2/3* and *nrc2/3/4* CRISPR lines (**Figs 1A**, **1C** and **S1B**), Cf-4/Avr4 expression resulted in chlorosis in the *nrc2/3* and *nrc2/3/4* lines which did not develop into confluent cell death (**Figs 1C** and **S1B**). These results indicate that the Cf-4/Avr4-triggered hypersensitive cell death requires NRC2/3.

## NRC3 mediates Cf-4/Avr4-triggered hypersensitive cell death in *N. benthamiana*

Next, we determined the individual contribution of NRC2 and NRC3 to Cf-4/Avr4-triggered cell death. To do this, we transiently complemented the *nrc2/3/4* CRISPR lines with either NRC2 or NRC3 in addition to expressing Cf-4/Avr4 (**Fig 2A**). As a control, we used Pto/AvrPto, which can utilize either NRC2 or NRC3 for Prf-mediated hypersensitive cell death signalling in *N. benthamiana* [41]. In addition to NRC2 and NRC3, we used NRC4 which is not expected to complement the Cf-4/Avr4 or Pto/AvrPto-triggered cell death in the *N. benthamiana nrc2/3/4* lines. Although it was previously reported that NRC1 is required for Cf-4/Avr4 hypersensitive cell death [38], *N. benthamiana* and other species in the *Nicotiana* genus lack *NRC1* (**S3 Fig**). Thus, we decided to use tomato NRC1 (*Sl*NRC1) to test whether *Sl*NRC1 can also complement Cf-4/Avr4-triggered cell death in *N. benthamiana*. All tested NRCs accumulate, however NRC3 accumulates to lower levels as compared to *Sl*NRC1, NRC2, or NRC4 (**Fig 2B**). In the *nrc2/3/4* lines, Cf-4/Avr4 and Pto/AvrPto-triggered cell death responses were complemented by NRC3 compared to the empty vector (EV) or NRC4 control (**Figs 2C** and **2D**, **S4A** for statistical analysis). NRC2 complemented Pto/AvrPto cell death in the *nrc2/3/4* lines, but only partially complemented Cf-4/Avr4-triggered cell death as compared to NRC3 (**Fig 2C** and **2D**). Finally, *Sl*NRC1 partially complemented Pto/AvrPto cell death, but not Cf-4/Avr4 cell death (**Figs 2C** and **2D**, **S4A** for statistical analysis). This difference is not due to differential protein accumulation, as NRC3 protein accumulates less than *Sl*NRC1 or NRC2 and NRC4 (**Fig 2C**). Whereas NRC1 is pseudogenized in the *Nicotiana* genus with only small fragments corresponding to the LRR being found in the genome assemblies, NRC2 and NRC3 are conserved in all Solanaceous species investigated (**S3 Fig**). Indeed, both tomato and pepper NRC3 (*Sl*NRC3 and *Ca*NRC3, respectively) can complement Cf-4/Avr4-triggered cell death in the *nrc2/3/4* lines, but not tomato or pepper NRC1 (*Sl*NRC1 and *Ca*NRC1) or pepper NRC2 (*Ca*NRC2) (**S5 Fig**). Tomato NRC2 (*Sl*NRC2) can partially complement Cf-4/Avr4-triggered cell death as compared to tomato NRC3 (**S5 Fig**). Furthermore, NRC3 can complement Cf-2/Rcr3/Avr2-, Cf-5/Avr5-, and Cf-9/Avr9-triggered cell death in the *nrc2/3/4* lines, while NRC2 can partially complement Cf-9/Avr9-triggered cell death as observed for Cf-4/Avr4 (**S6 Fig**). We conclude that NRC3 is the main NRC helper node contributing to cell-surface receptor-triggered hypersensitive cell death in the Solanaceae.

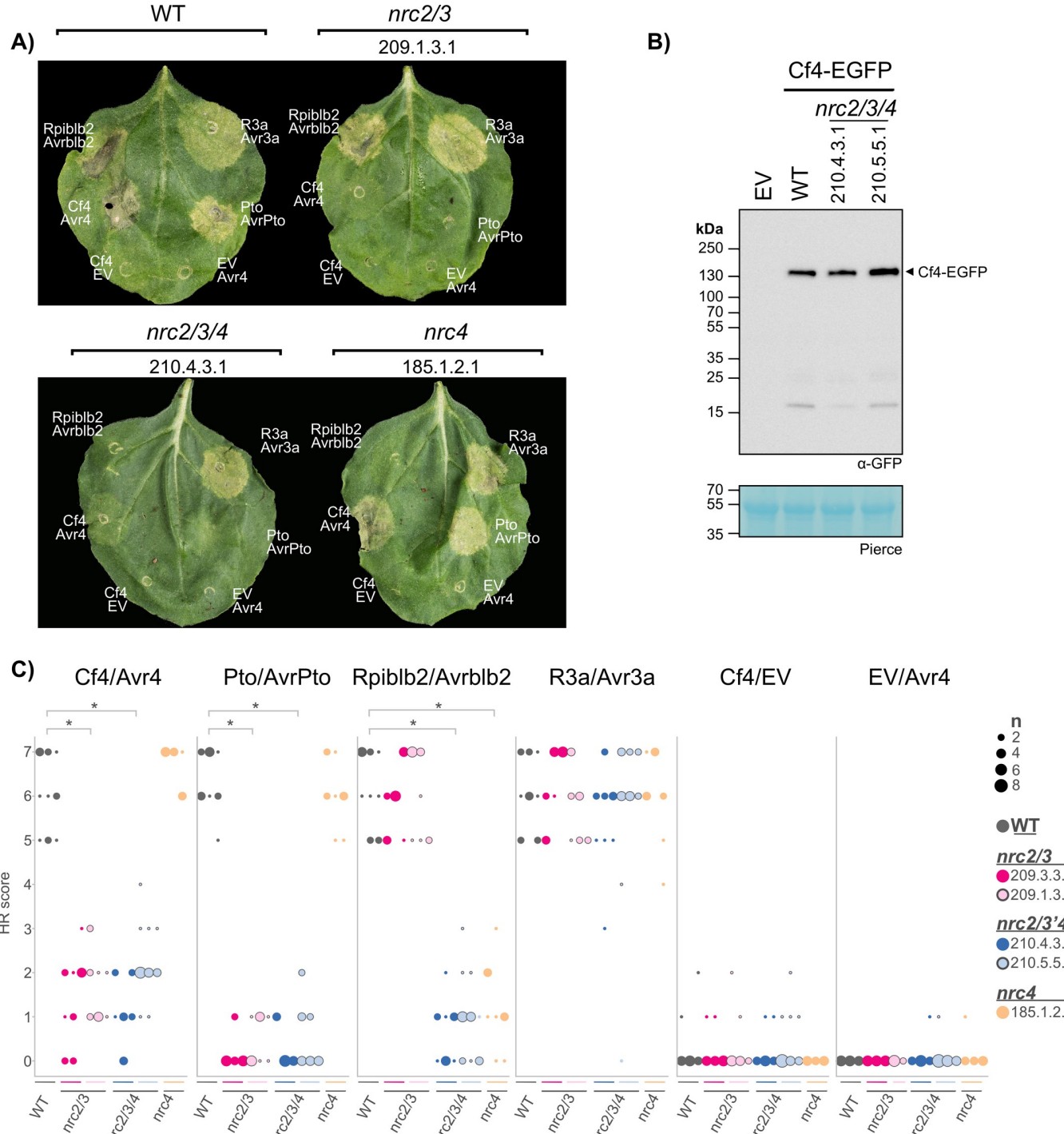

**Fig 1. Cf-4/Avr4-triggered hypersensitive cell death requires NRC2/3. A)** Cf-4/Avr4-triggered hypersensitive cell death is lost in the *nrc2/3* and *nrc2/3/4* knock-out lines, but not in the *nrc4* line. Representative *N. benthamiana* leaves infiltrated with appropriate constructs photographed 7–10 days after infiltration. The NRC CRISPR lines, *nrc2/3*-209.1.3.1, *nrc2/3/4*-210.4.3.1, and *nrc4*-185.1.2.1, are labelled above the leaf and the receptor/effector pair tested, Cf-4/Avr4, Prf (Pto/AvrPto), Rpi-blb2/AVRblb2 or R3a/Avr3a, are labelled on the leaf image. Cf-4/EV and EV/Avr4 were also included as negative controls. A representative leaf of the independent *nrc2/3*-209.3.3.1, and *nrc2/3/4*-210.5.5.1 CRISPR line are shown in **S1B Fig**) Cf-4 protein accumulation is not affected in the *nrc2/3/4* lines. For the immunoblot analysis, total protein was extracted 5 days after transient expression of Cf-4-EGFP by agroinfiltration in wild-type, *nrc2/3/4*-210.4.3.1 and *nrc2/3/4*-210.5.5.1 *N. benthamiana* leaves. Cf-4-EGFP accumulation was detected using anti-GFP antibody. **C)** Quantification of hypersensitive cell death. Cell death was scored based on a 0–7 scale (**S1 Fig**) between 7–10 days post infiltration. The results are presented as a dot plot, where the size of each dot is proportional to the count of the number of samples with the same score within each biological replicate. The experiment was

independently repeated three times. The columns correspond to the different biological replicates. Significant differences between the conditions are indicated with an asterisk (*). Details of statistical analysis are presented in **S1 Fig**.

## The NRC3 N-terminal MADA motif is required for Cf-4-triggered hypersensitive cell death

The NRC-helper clade is characterized by a N-terminal MADA-type α1 helix diagnostic of ZAR1-type CC-NLRs [46]. In order to determine whether the N-terminal MADA motif of NRC3 is required for the Cf-4/Avr4-triggered hypersensitive cell death, we took advantage of recently described cell death abolishing "MADA" mutations in this motif [46]. Like ZAR1 and NRC4, both NRC2 and NRC3 have an N-terminal MADA-type α1 helix (**Fig 3A**). Mutating leucine 17 to glutamic acid (L17E) abolishes NRC4-mediated cell death and immunity [46]. L17E and L21E mutations in NRC2 (NRC2$^{L17E}$) and NRC3 (NRC3$^{L21E}$) correspond to the L17E mutation in NRC4 (NRC4$^{L17E}$) (**Fig 3A** and **3B**). To establish whether these mutations affect NRC2- or NRC3-mediated cell death, we conducted a complementation assay of Cf-4/Avr4 or Pto/AvrPto cell death in *N. benthamiana nrc2/3/4* lines with either wild-type NRC2 or NRC3, or their respective MADA mutants (**Fig 3B**). The MADA mutation did not affect protein accumulation, as both NRC2, NRC3, and their respective MADA mutants accumulated to similar levels (**Fig 3C**). Unlike wild-type NRC3, NRC3$^{L21E}$ didn't complement either Cf-4/Avr4 or Pto/AvrPto-triggered cell death in the *nrc2/3/4* lines (**Fig 3D** and **3E**). As previously noted, NRC2 only partially complemented Cf-4/Avr4-triggered cell-death as compared to NRC3 (**Figs 2** and **S4**), but the NRC2$^{L17E}$ failed to complement either Cf-4/Avr4-triggered cell-death or Pto/AvrPto-triggered cell death (**Figs 3D** and **3E**, **S7A** for the statistical analysis). We conclude that the N-terminal α1 helix of NRC2 and NRC3 is required for their function as helper-NLRs downstream of Cf-4 activation.

## A chimeric protein fusion of the Arabidopsis ZAR1 α1 helix to NRC2/3 can complement loss of Cf-4-triggered hypersensitive cell death

In the activated ZAR1 resistosome, the N-terminal α1 helix of the CC domain is an important determinant for cell death induction [7]. In order to determine whether the N-terminal α1 helix of the ZAR1 CC domain could substitute for the NRC2/NRC3 predicted α1 helix in Cf-4/Avr4-triggered hypersensitive cell death, we generated chimaeras of NRC2 or NRC3 (NRC2$^{ZAR1\alpha1}$ and NRC3$^{ZAR1\alpha1}$), where the first 17 amino acids of ZAR1 corresponding to the α1 helix (**Fig 4A**) substitute the first 17 amino acids of NRC2 or the first 21 amino acids of NRC3. To test whether these chimaeras can complement Cf-4/Avr4 or Pto/AvrPto-triggered cell death, we transiently expressed these receptor/effector pairs with either wild-type NRC2 or NRC3, or their respective ZAR1 α1 helix chimaeras in *N. benthamiana nrc2/3/4* lines (**Fig 4B**). The chimeric proteins expressed and accumulated to similar level as both wild-type NRC2 and NRC3 proteins (**Fig 4C**). Like wild-type NRC3, NRC3$^{ZAR1\alpha1}$ complemented both Cf-4/Avr4 or Pto/AvrPto-triggered cell death in the *N. benthamiana nrc2/3/4* CRISPR lines (**Fig 4D** and **4E**). Surprisingly, the NRC2$^{ZAR1\alpha1}$ chimaera complemented Cf-4/Avr4-triggered cell death to a similar extent as NRC3 (**Fig 4D and 4E**). The NRC2$^{ZAR1\alpha1}$ chimaera also complemented Pto/AvrPto-triggered cell death to a greater extent than wild-type NRC2 (**Figs 4D** and **4E**, **S8A** for the statistical analysis). Importantly, this was not because the NRC ZAR1 α1 helix chimaeras are autoactive, as expressing them with either Cf-4 or Pto and an EV control did not result in cell death (**Fig 4D and 4E**). Furthermore, while NRC3 with the NRC2 α1 helix (NRC3$^{NRC2\alpha1}$) can complement Cf-4/Avr4- and Pto/AvrPto-triggered cell death, NRC2 with the NRC3 α1 helix (NRC2$^{NRC3\alpha1}$) cannot (**S9 Fig**). In ZAR1 E11 is required to generate a

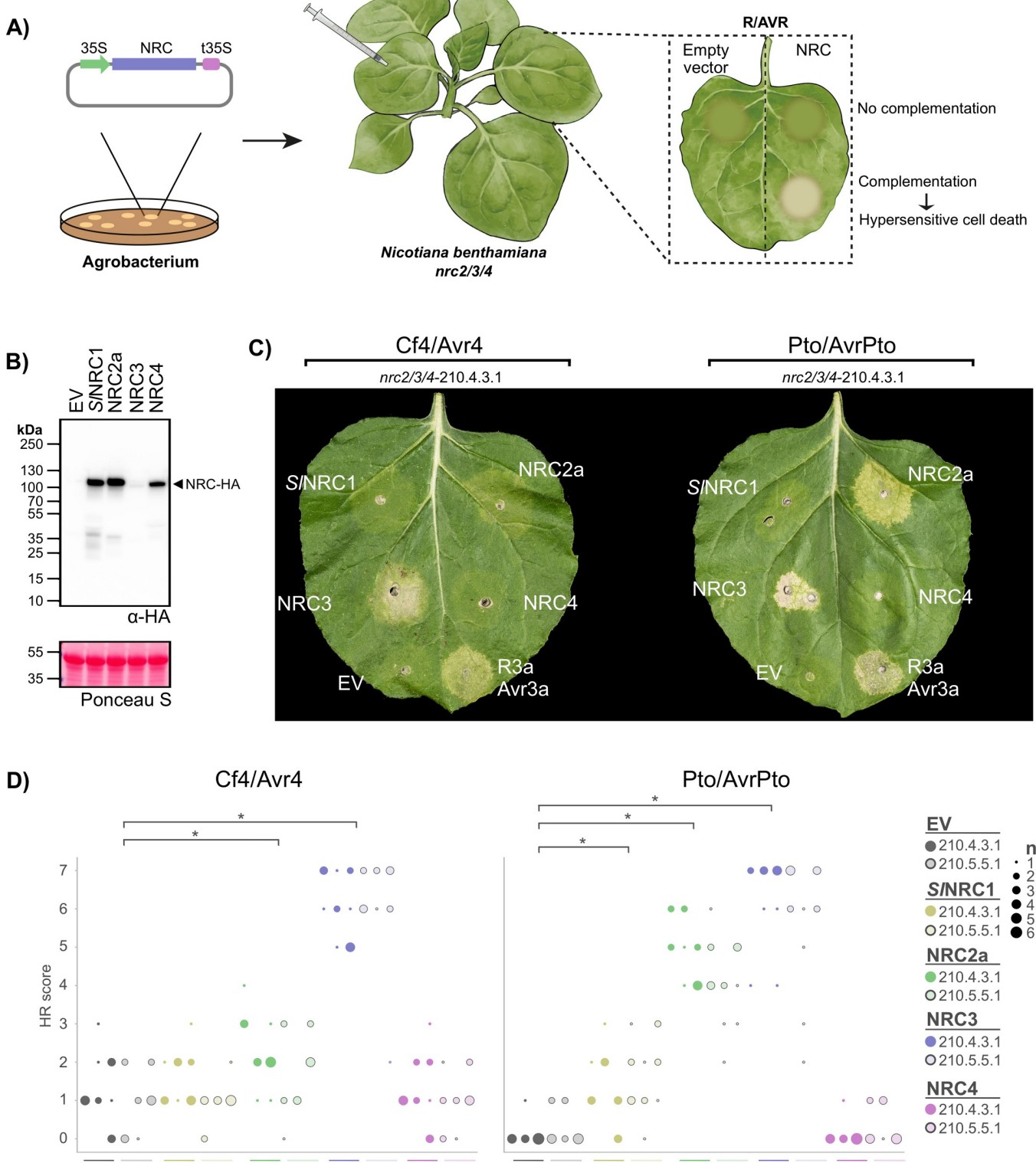

**Fig 2. NRC3 complements Cf-4/Avr4-triggered hypersensitive cell death in the *nrc2/3/4* knock-out lines. A)** A schematic representation of the cell death complementation assay. NRCs and empty vector (EV) were transformed into *A. tumefaciens*, and transiently co-expressed *in N. benthamiana* plants with either

Cf-4/Avr4 or Pto/AvrPto. Hypersensitive cell death was scored based on a modified 0–7 scale between 7–10 days post infiltration (**S1A Fig**). **B**) *Sl*NRC1, NRC2, NRC3, and NRC4 protein accumulation. Immunoblot analysis of transiently expressed C-terminally 6xHA-tagged NRCs 5 days after agroinfiltration in wild-type *N. benthamiana* plants. **C**) Cf-4/Avr4-triggered hypersensitive cell death is complemented in the *nrc2/3/4* lines with NRC3 and to a lesser extent NRC2, but not *Sl*NRC1 or NRC4. Representative *N. benthamiana* leaves infiltrated with appropriate constructs were photographed 7–10 days after agroinfiltration. The receptor/effector pair tested, Cf-4/Avr4 and Prf (Pto/AvrPto), are labelled above the leaf of *nrc2/3/4*-210.4.3.1. The NRC used for complementation or EV control are labelled on the leaf image. A representative leaf of the independent *nrc2/3/4*-210.5.5.1 line is shown in **S4D Fig**) Quantification of hypersensitive cell death. Cell death was scored based on a 0–7 scale between 7–10 days post infiltration. The results are presented as a dot plot, where the size of each dot is proportional to the count of the number of samples with the same score within each biological replicate. The experiment was independently repeated three times. The columns correspond to the different biological replicates. Significant differences between the conditions are indicated with an asterisk (*). Details of statistical analysis are presented in **S4 Fig**.

$Ca^{2+}$-permeable membrane-permeable pore and induce cell death [7]. However, this residue is not required for the wheat NLR Sr35 to generate a $Ca^{2+}$-permeable membrane-permeable pore and induce cell death [8], and is similarly also not required for NRC4-mediated cell-death [46]. Mutating the equivalent glutamic acid residue in NRC3 to glutamine (E14Q) does not impair complementation of Cf-4/Avr4- or Pto/AvrPto-triggered cell death in the *nrc2/3/4* CRISPR lines (**S10 Fig**). We conclude that the α1 helix of ZAR1 can replace the NRC2 and NRC3 α1 helix for hypersensitive cell death signaling, and even enhances the NRC-dependent hypersensitive cell-death response. Why the NRC2^ZAR1α1 chimaera, but not the NRC2^NRC3α1 chimaera, can complement Cf-4/Avr4-triggered cell death to a similar extent as NRC3 is not clear. We conclude that modifications of the α1 helix of NRCs can sometimes expand their role as helper-NLRs.

## Pathogen effectors can suppress Cf4-triggered hypersensitive cell death in a NRC3-dependent manner

We recently showed that plant pathogens have converged on suppressing the NRC network to prevent immune responses [47]. NRC3, for example, is suppressed by both the cyst nematode *Globodera rostochiensis* effector SPRYSEC15, as well as the oomycete *Phytophthora infestans* effector AVRcap1b [47]. Since Cf-4/Avr4-triggered cell death requires NRC3, we reasoned that these effectors could also suppress cell death triggered by Cf-4 and similar LRR-RPs. To test this hypothesis, we generated heterologous expression constructs containing both NRC3 and either SPRYSEC15, AVRcap1b, or mCherry (control). We then co-expressed these NRC3/suppressor combinations together with Cf-4/Avr4 in *N. benthamiana nrc2/3/4* lines (**Fig 5A**). As controls, we used either autoactive NRC3^D480V, or the receptor/effector pair Pto/AvrPto, both of which are suppressed by SPRYSEC15 and AVRcap1b [47]. The receptor/effector pair R3a/Avr3a, which is not affected by SPRYSEC15 or AVRcap1b, was used as a control combination [47]. These experiments revealed that both AVRcap1b and SPRYSEC15 suppress Cf-4/Avr4-triggered cell death (**Figs 5B and 5C**, **S11A** for the statistical analysis). As expected, AVRcap1b and SPRYSEC15 suppressed autoactive NRC3^D480V and Pto/AvrPto-triggered cell death, but not R3a/Avr3a-triggered cell death (**Fig 5B and 5C**). As previously noted, transient expression of Cf-4/Avr4 results in a NRC3-independent chlorosis (**Fig 1**), and this phenotype was not suppressed by either SPRYSEC15 or AVRcap1b (**Fig 5B and 5C**). We conclude that divergent pathogen effectors, acting as suppressors of NRC3-mediated cell death, can suppress the hypersensitive cell death induced by the cell-surface receptor Cf-4.

## Discussion

Cell surface receptor-mediated immunity and intracellular NLR-mediated immunity are viewed as the main innate immune responses in plants. Although their signaling pathways were initially thought to be qualitatively distinct, recent findings point to a degree of cross-talk

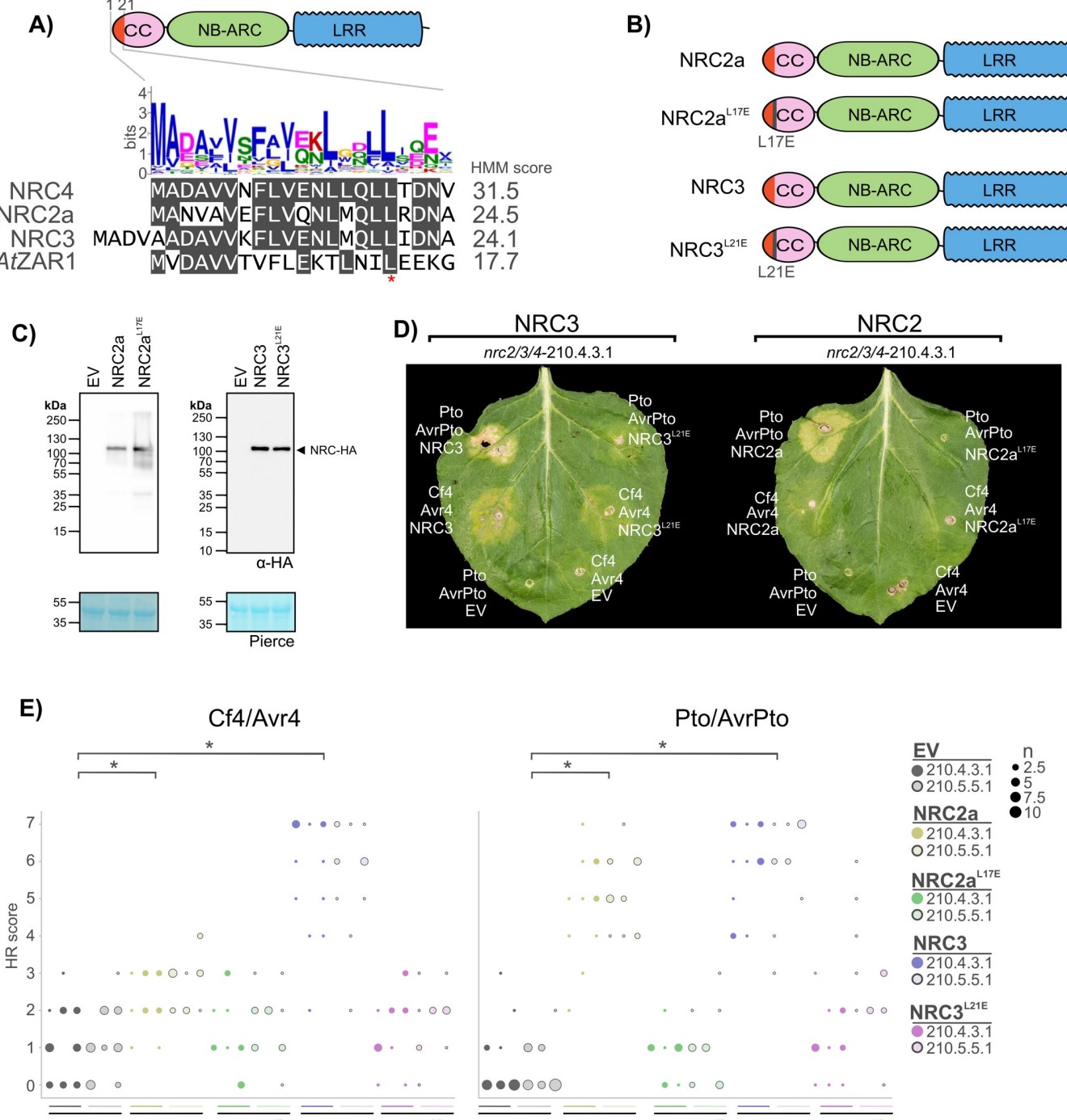

**Fig 3. The N-terminal MADA motif of NRC3 is required for Cf-4/Avr4-triggered hypersensitive cell death. A**) NRC3 has an N-terminal MADA motif. Alignment of NRC2, NRC3, NRC4 and ZAR1 N-terminal MADA motif with the consensus sequence pattern of the MADA motif and the HMM score for each sequence. **B**) Schematic representation of NRC2, NRC3, and the respective NRC2[L17E] and NRC3[L21E] MADA mutants. **C**) NRC2 and NRC3 MADA mutants accumulate to similar levels as wild-type proteins. Immunoblot analysis of transiently expressed C-terminally 6xHA-tagged NRCs 5 days after agroinfiltration in wild-type *N. benthamiana* plants. **D**) NRC MADA mutants do not complement Cf-4/Avr4 or Pto/AvrPto-triggered hypersensitive cell death in the *N. benthamiana nrc2/3/4* lines. Representative *N. benthamiana* leaves infiltrated with appropriate constructs were photographed 7–10 days after infiltration. The NRCs tested, NRC2 and NRC3, are labelled above the leaf of *nrc2/3/4*-210.4.3.1. The receptor/effector pair tested, Cf-4/Avr4 and Prf (Pto/AvrPto), are labelled on the leaf image. A representative leaf of the independent *nrc2/3/4*-210.5.5.1 line is shown in **S7E Fig**) Quantification of hypersensitive cell death. Cell death was scored based on a 0–7 scale between 7–10 days post infiltration. The results are presented as a dot plot, where the size of each dot is proportional to the count of the number of samples with the same score within each biological replicate. The experiment was independently repeated three times. The columns

correspond to the different biological replicates. Significant differences between the conditions are indicated with an asterisk (*). Details of statistical analysis are presented in S7 Fig.

[30–32,37]. Here we show that the cell-surface leucine-rich repeat receptor-like protein Cf-4 taps into the NRC helper NLR network to trigger hypersensitive cell death upon recognition of the apoplastic fungal *Cladosporium fulvum* effector Avr4. We propose that NRC3 is a core node that connects cell surface and intracellular immune networks, which may involve uncharacterized sensor-NLRs (**Fig 6**). Divergent plant pathogens have convergently evolved to target the NRC3 node, thereby suppressing both intracellular and cell surface triggered immune responses (**Fig 6**).

The hypersensitive cell death elicited by Cf-4/Avr4 is abolished in *N. benthamiana nrc2/3* and *nrc2/3/4*, but not *nrc4* lines (**Fig 1**). Interestingly, transient expression of Cf-4/Avr4 triggers a NRC2/3/4-independent chlorosis (**Fig 1**) indicating that there are probably both NRC-dependent and -independent responses triggered by Cf-4, possibly mediated by the complex and dynamic association of RPs such as Cf-4 with partner RKs and RLCKs [21]. Although complementation using *N. benthamiana* or tomato NRC2, but not pepper NRC2, only partially rescues the phenotype, complementation using *N. benthamiana*, tomato, and pepper NRC3 significantly restores Cf-4/Avr4-triggered hypersensitive cell death in the *nrc2/3* and *nrc2/3/4* CRISPR lines (**Fig 2**). Since NRC3 is conserved in all solanaceous species investigated (**S3 Fig**) it may mediate cell death responses triggered by cell-surface receptors in these species as well. When in evolutionary time this member of the NRC helper NLR family evolved to connect to the signaling pathway triggered by pattern recognition receptors remains to be elucidated.

Previously, NRC1 was implicated in the cell death mediated by the cell surface receptors Cf-4 [38], LeEIX2 [39] and Ve1 [40] in *N. benthamiana*. Here we show that NRC3, rather than NRC1, links cell-surface immune receptors to intracellular immunity in the *N. benthamiana* experimental system (**Figs 1** and **2**). Additionally, *Sl*NRC1 cannot complement Cf-4/Avr4-triggered hypersensitive cell death in the *N. benthamiana* experimental system (**Fig 2**). Unlike *Sl*NRC1, NRC3 is conserved in all Solanaceous species (**S3 Fig**), indicating that this link between cell surface immune receptors and intracellular immunity may be conserved in the Solanaceae. In addition to the NRC-dependent hypersensitive cell death, NRC-independent responses are also activated by these cell surface immune receptors. It may be that for some pathogens the NRC-independent response is sufficient to result in immunity, while for other pathogens the NRC-dependent response is also required. Which genetic components are involved in the NRC-independent response downstream of cell surface receptors remains a topic of further study.

NRC3 requires an intact MADA motif to mediate Cf-4/Avr4-triggered hypersensitive cell death (**Fig 3**). This is similar to intracellular activation of NRCs, where NRC4-mediated Rpi-blb2/AVRblb2-triggered cell death and immunity require an intact NRC4 MADA motif [46]. Additionally, the ZAR1 α1 helix can functionally substitute for the N-terminal equivalent sequence of NRC3. This observation suggests that the hypersensitive cell death response downstream of Cf-4/Avr4 immune signaling could involve an activated resistosome. Indeed, NRC2 was recently shown to oligomerize into a resistosome-like structure upon activation by multiple sensor NLRs, indicating that NRC-type helper NLRs are likely to function like ZAR1 [53,54].

Surprisingly, swapping the ZAR1 α1 helix into NRC2 resulted in a stronger cell death response as compared to the native NRC2. In contrast to the native NRC2, this chimeric protein could efficiently complement Cf-4/Avr4-triggered hypersensitive cell death (**Fig 4**). Given

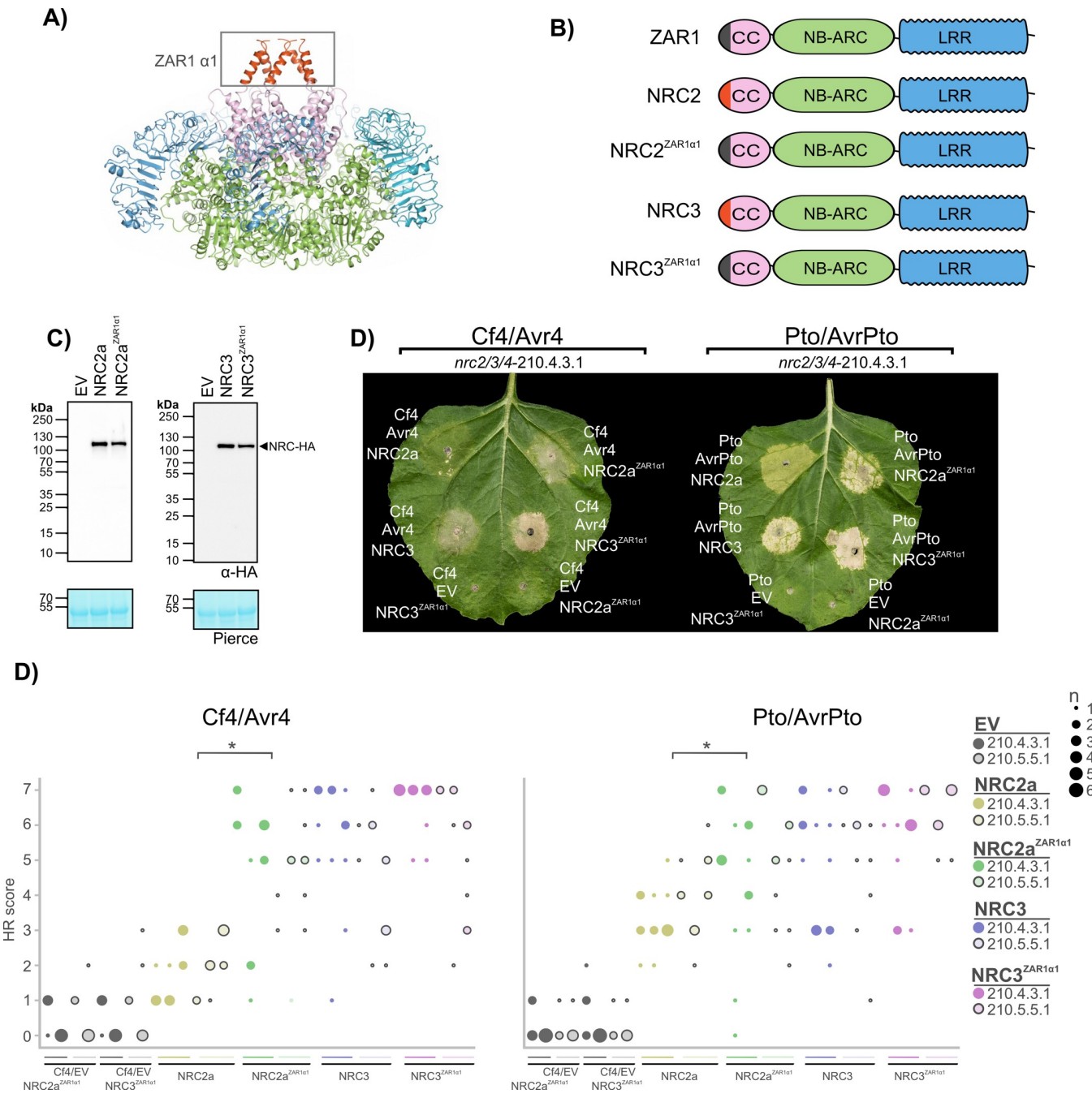

**Fig 4. The ZAR1 α1 helix can functionally replace the NRC2 and NRC3 α1 helix for Cf-4/Avr4-triggered hypersensitive cell death. A)** Structure of the ZAR1 resistosome with the N-terminal α1 helix highlighted. **B)** Schematic representation of ZAR1, NRC2, NRC3, and the respective NRC2[ZAR1α1] and NRC3[ZAR1α1] chimaeras in which residues 1–17 and 1–21 are replaced by residues 1–17 from ZAR1, respectively. **C)** NRC ZAR1 α1 helix chimaeras accumulate to similar levels as wild-type NRC proteins. Immunoblot analysis of transient NRC-6xHA accumulation 5 days after agroinfiltration in wild-type *N. benthamiana* plants. **D)** NRC ZAR1 α1 helix chimaeras can complement Cf-4/Avr4 and Pto/AvrPto-triggered hypersensitive cell death in the *N. benthamiana nrc2/3/4* CRISPR lines. Representative *N. benthamiana* leaves infiltrated with appropriate constructs were photographed 7–10 days after infiltration. The receptor/effector pair tested, Cf-4/Avr4 and Prf (Pto/AvrPto), are labelled above the leaf of NRC CRISPR line *nrc2/3/4*-210.4.3.1. The NRC tested, NRC2 and NRC3, are labelled on the leaf image. To ensure the NRC ZAR1 α1 helix chimaeras were not autoactive when expressed with either Cf-4 or Pto an EV control was taken along. A representative leaf of the independent *nrc2/3/4*-210.5.5.1 CRISPR line is shown in **S8E Fig**) Quantification of hypersensitive cell death. Cell death was scored based on a 0–7 scale between 7–10 days post infiltration. The results are presented as a dot plot, where the size of each dot is proportional to the count of the number of samples with the same score within each biological replicate. The experiment was independently repeated three times. The columns correspond to the different biological replicates. Significant differences between the conditions are indicated with an asterisk (*). Details of statistical analysis are presented in **S8 Fig**.

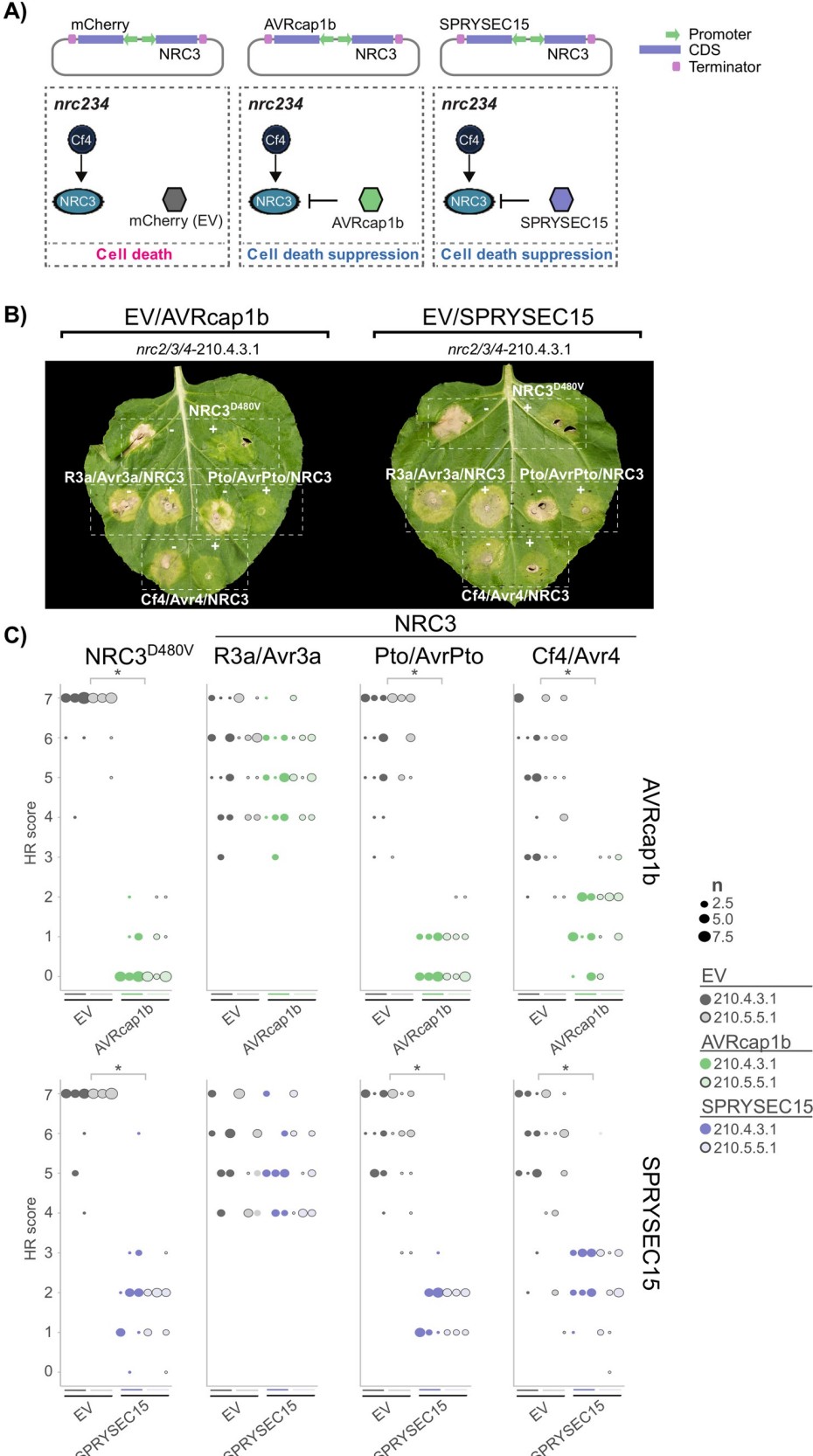

**Fig 5. Cf-4/Avr4-triggered hypersensitive cell death can be suppressed by divergent pathogen effectors. A)** A schematic representation of cell death suppression assay. *A. tumefaciens* containing vectors carrying both NRC3 and either SPRYSEC15, AVRcap1b, or mCherry as empty vector control, were co-expressed in *N. benthamiana nrc2/3/4* CRISPR lines, with the effectors suppressing the cell death response. **B)** Cf-4/Avr4-triggered hypersensitive cell death is suppressed by SPRYSEC15 and AVRcap1b, but not the EV control. Representative image of *N. benthamiana nrc2/3/4-*210.4.3.1 CRISPR line leaves which were agroinfiltrated with NRC3/suppressor constructs, as indicated above the leaf, and either autoactive NRC3$^{D480V}$, Prf (Pto/AvrPto), R3a/Avr3a, or Cf-4/Avr4 as labelled on the leaf image, photographed 7–10 days after infiltration. A representative leaf of the independent *nrc2/3/4-*210.5.5.1 CRISPR line is shown in **S11C Fig) C)** Quantification of hypersensitive cell death. Cell death was scored based on a 0–7 scale between 7–10 days post infiltration. The results are presented as a dot plot, where the size of each dot is proportional to the count of the number of samples with the same score within each biological replicate. The experiment was independently repeated three times. The columns correspond to the different biological replicates. Significant differences between the conditions are indicated with an asterisk (*). Details of statistical analysis are presented in **S11 Fig**.

that the stronger immune responses triggered by the NRC2$^{ZAR1α1}$ chimaera are not due to enhanced protein accumulation, the specific mechanisms that explain these differences remain unclear (**Fig 4**). It may be that this chimeric protein is more "trigger-happy", either due to reduced autoinhibition or enhanced capacity to generate the resistosome funnel. Often trigger-happy mutations give increased autoimmune responses [55–57]. However, we did not observe autoimmune responses with the NRC2$^{ZAR1α1}$ chimaera. Alternatively, the ZAR1 α1 helix may stabilize activated NRC2. It may also be involved in subcellular localization and targeting either prior to, or after activation, and this localization could affect the strength of the immune response. Regardless of the mechanism, this chimeric approach has biotechnological implications and may be useful in engineering trigger-happy helper NLRs, such as the NRCs, that can mediate more potent immune responses.

The recent observation that the ZAR1 α1 helix forms a resistosome that acts as a Ca$^{2+}$-permeable membrane-permeable pore [7] raises the possibility that a similar mechanism mediates Cf-4/Avr4-triggered cell death. This signaling could be either a cell autonomous response directly downstream of activated cell surface receptors, or a response involved in amplification of the initial signaling response in either a cell autonomous or non-autonomous fashion. Genetic dissection of the NRC-independent responses may yield answers to these questions. It could very well be that upon activation of PRRs by pathogen molecules, multiple Ca$^{2+}$ channels are activated during various stages in the signaling cascade, thereby explaining previous difficulties in genetically dissecting this response [9,32,58–60].

We recently discovered that two effectors from unrelated pathogens suppress NRC3-mediated immune responses [47]. Here, we found that these two effectors can also suppress cell surface immune signaling through acting on NRC3 (**Fig 6**). In the case of SPRYSEC15 this suppression involves direct binding of the effector to the NB-ARC domain of NRC proteins, while the suppression mediated by AVRcap1b is indirect [47]. Therefore, in addition to suppressing intracellular immune activation, SPRYSEC15 and AVRcap1b can also suppress immune activation triggered by cell-surface receptors. Perhaps this is a more common approach by pathogens. We hypothesize that various cell surface receptors that are involved in nematode and oomycete immunity would also be dependent on helper NLRs of the NRC family similar to the fungal resistance protein Cf-4.

We have previously shown that the leucine-rich repeat receptor-like kinase FLS2 does not require NRC2/3/4 for triggering MAPK phosphorylation and a ROS burst upon recognition of the bacterial flagellin epitope flg22 [48]. Furthermore, Suppressor of the G2 allele of skp1 (SGT1)-a core component of the CC-NLR-mediated hypersensitive cell death-is not required for receptor-like kinase signaling in *N. benthamiana* or Arabidopsis [61], further confirming that NRCs are not always downstream of leucine-rich repeat receptor kinases. It therefore

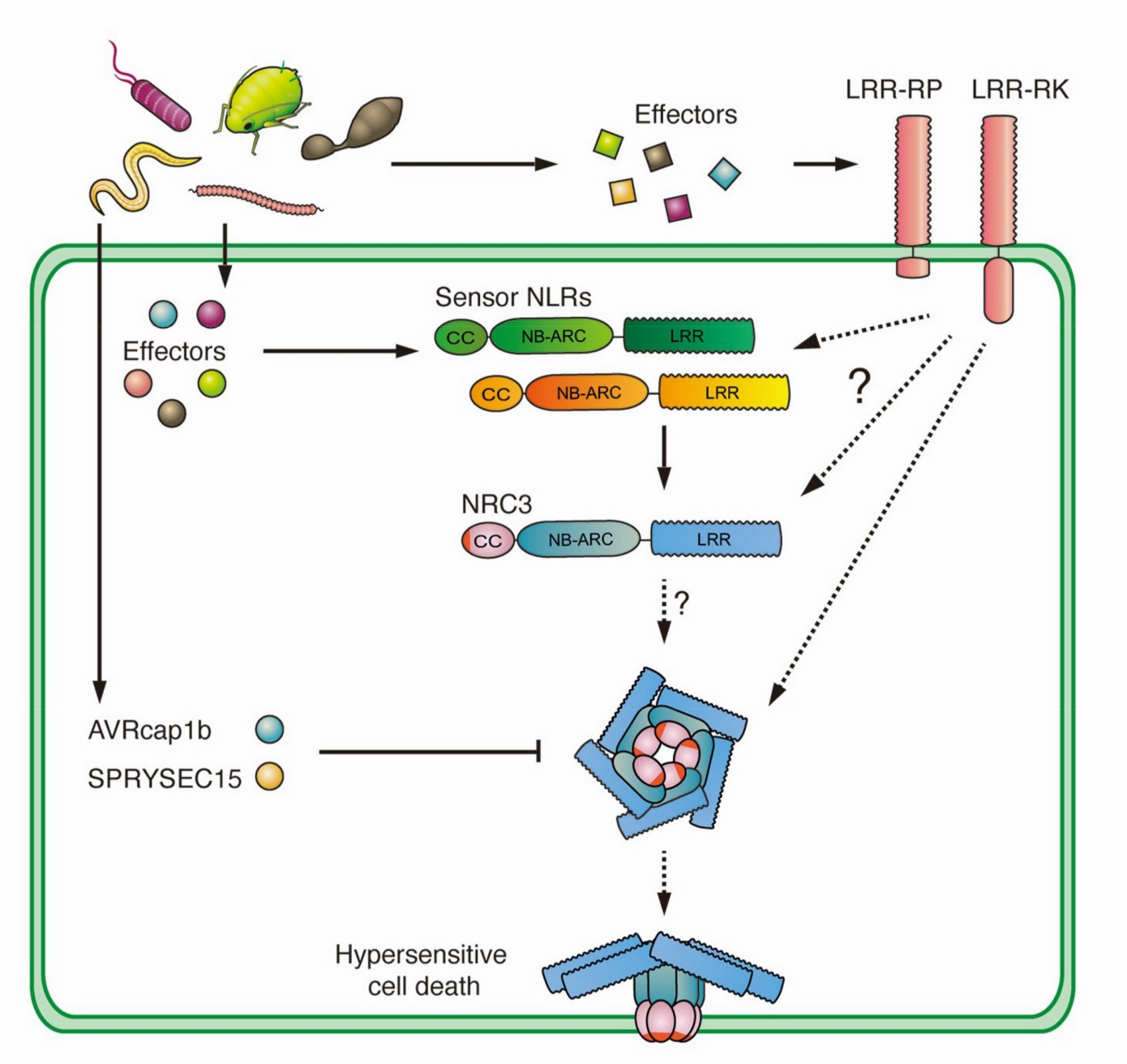

**Fig 6. Hypersensitive cell death triggered by activation of cell-surface immune receptors requires the NRC network in Solanaceae.** Recognition of apoplastic effectors and translocated intracellular effectors by cell-surface leucine-rich repeat receptor proteins and intracellular NRC-type sensor-NLRs, respectively, results in NRC3-mediated hypersensitive cell death. This cell death requires an intact α1 helix of NRC3 indicating the possible formation of an activated NRC3 resistosome. Divergent pathogen effectors converge on this node to suppress both NLR triggered as well as cell-surface receptor triggered immune recognition.

seems likely that NRC3 is only required for full activation of the subset of cell surface receptors that can trigger hypersensitive cell death. Since NRCs are transcriptionally upregulated upon activation of Cf-4, FLS2 and other cell surface immune receptors [38] it may be that Cf-4 activation results in transcriptional upregulation of NRC3 in a NRC-independent manner, to

amplify the immune response and trigger hypersensitive cell death in a second phase in a NRC3-dependent manner.

Intracellular NLR immune receptors were recently shown to require intact cell surface immune signaling in order to efficiently trigger hypersensitive cell death [30,31]. However, the mechanistic links between cell surface immune signaling and the activation of intracellular immune receptors remain unclear [33]. Here we show that intracellular NLR immune receptors are also required for hypersensitive cell death induction by a cell surface receptor. In addition, we show that this hypersensitive cell death likely requires activation of a MADA-CC-NLR type resistosome. Understanding how these signaling modules intertwine and have evolved and how pathogens suppress these responses can be leveraged to guide new approaches for breeding disease resistance.

## Material and methods

### Plant material and growth conditions

Wild-type and mutant *N. benthamiana* were propagated in a glasshouse and, for most experiments, were grown in a controlled growth chamber with temperature 22–25˚C, humidity 45–65% and 16/8 hr light/dark cycle. The CRISPR lines used have been previously described: *nrc2/3*-209.1.3.1 and *nrc2/3*-209.3.3.1 [49], *nrc2/3/4*-210.4.3.1 and *nrc2/3/4*-210.5.5.1 [48], *nrc4*-185.1.2.1 [46], and *eds1* [62].

### Plasmid constructions

The Golden Gate Modular Cloning (MoClo) kit [63] and the MoClo plant parts kit [64] were used for cloning, and all vectors are from this kit unless specified otherwise. Effectors, receptors and NRCs were cloned into the binary vector pJK268c, which contains the Tomato bushy stunt virus silencing inhibitor p19 in the backbone [65]. Cloning design and sequence analysis were done using Geneious Prime (v2021.2.2; https://www.geneious.com). Plasmid construction is described in **S1 Table**.

### Transient gene-expression and cell death assays

Transient gene expression in *N. benthamiana* were performed by agroinfiltration according to methods described by van der Hoorn *et al.*, [66]. Briefly, *A. tumefaciens* strain GV3101 pMP90 carrying binary vectors were inoculated from glycerol stock in LB supplemented with appropriate antibiotics and grown O/N at 28˚C until saturation. Cells were harvested by centrifugation at $2000 \times g$, RT for 5 min. Cells were resuspended in infiltration buffer (10 mM $MgCl_2$, 10 mM MES-KOH pH 5.6, 200 μM acetosyringone) to the appropriate $OD_{600}$ (see **S1 Table**) in the stated combinations and left to incubate in the dark for 2 h at RT prior to infiltration into 5-week-old *N. benthamiana* leaves. HR cell death phenotypes were scored in a range from 0 (no visible necrosis) to 7 (fully confluent necrosis) according to Adachi *et al.*, [46] (**S1A Fig**).

### Protein immunoblotting

Six *N. benthamiana* leaf discs (8 mm diameter) taken 5 days post agroinfiltration were homogenised in extraction buffer [10% glycerol, 25 mM Tris-HCl, pH 7.5, 1 mM EDTA, 150 mM NaCl, 1% (w/v) PVPP, 10 mM DTT, 1x protease inhibitor cocktail (SIGMA), 0.2% IGEPAL CA-630 (SIGMA)]. The supernatant obtained after centrifugation at 12,000 x g for 10 min 4 ˚C was used for SDS-PAGE. Immunoblotting was performed with rat monoclonal anti-HA antibody (3F10, Roche) or mouse monoclonal anti-GFP antibody conjugated to HRP (B-2, Santa Cruz Biotech) in a 1:5000 dilution. Equal loading was validated by staining the PVDF

membranes with Pierce Reversible Protein Stain Kit (#24585, Thermo Fisher) or using Ponceau S (SIGMA).

## Bioinformatic and phylogenetic analyses of the NRC-helper family

NRC sequences were retrieved by BLASTP against the NCBI RefSeq proteomes using the tomato and *N. benthamiana* NRC proteins (*Sl*NRC1, *Sl*NRC2, *Sl*NRC3, *Sl*NRC4a, *Sl*NRC4b, *Nb*NRC2a, *Nb*NRC2b, *Nb*NRC3, *Nb*NRC4a, *Nb*NRC4b, *Nb*NRC4c) as a query. A single representative splice-variant was selected per gene. Full-length amino acid sequences were aligned using MAFFT (v7.450; [67]) (**S1 dataset**). FastTree (v2.1.11; [68]) was used to produce a phylogeny of the NRCs which were rooted on XP_00428175 (**S2 Dataset**). The NRC phylogeny was edited using the iTOL suite (6.3; [69]).

## Supporting information

**S1 Fig. Statistical analysis of Cf-4/Avr4-triggered cell death in different *N. benthamiana* *nrc* mutant lines. A**) Cf-4/Avr4-triggered cell death was scored on a scale of 0–7, with 0 being no response, and 7 being fully confluent cell-death in the entire infiltrated sector. Visible cell death starts appearing at a score of 4. **B**) Statistical analysis was conducted using the besthr R package. The dots represent the ranked data and their corresponding means (dashed lines), with the size of each dot proportional to the number of observations for each specific value (count key below each panel). The panels on the right show the distribution of 100 bootstrap sample rank means, where the blue areas under the curve illustrate the 0.025 and 0.975 percentiles of the distribution. A difference is considered significant if the ranked mean for a given condition falls within or beyond the blue percentile of the mean distribution of the wild-type control. **C**) Cf-4/Avr4-triggered hypersensitive cell death is lost in the *nrc2/3* and *nrc2/3/4* CRISPR lines. Representative *N. benthamiana* leaves infiltrated with appropriate constructs were photographed 7–10 days after infiltration. The NRC CRISPR lines, *nrc2/3*-209.3.3.1, *nrc2/3/4*-210.5.5.1, are labelled above the leaf and the receptor/effector pair tested, Cf-4/Avr4, Prf (Pto/AvrPto), Rpi-blb2/AVRblb2 or R3a/Avr3a, are labelled on the leaf image. Cf-4/EV and EV/Avr4 were also included.
(TIF)

**S2 Fig. Cf-4/Avr4-triggered hypersensitive cell death does not require EDS1. A**) Cf-4/Avr4-triggered hypersensitive cell death is lost in the *nrc2/3/4* knock-out lines, but not in the *eds1* line. Representative *N. benthamiana* leaves infiltrated with appropriate constructs photographed 7–10 days after infiltration. The CRISPR lines, *nrc2/3/4*-210.4.3.1, and *eds1*, are labelled above the leaf and the receptor/effector pair tested, Cf-4/Avr4 (NRC2/3-depedent), Rx/CP (NRC2/3/4-dependent) or XopQ (Roq1, EDS1-dependent), are labelled on the leaf image. As noticed previously, transient XopQ expression gives a mild chlorotic response which is absent in the *eds1* CRISPR line [70]. Cf-4/EV and EV/Avr4 were also included as negative controls. **B**) Quantification of hypersensitive cell death. Cell death was scored based on a 0–7 scale (**S1 Fig**) between 7–10 days post infiltration. The results are presented as a dot plot, where the size of each dot is proportional to the count of the number of samples with the same score within each biological replicate. The experiment was independently repeated three times. The columns correspond to the different biological replicates. Significant differences between the conditions are indicated with an asterisk (*). **C**) Statistical analysis was conducted using the besthr R package. The dots represent the ranked data and their corresponding means (dashed lines), with the size of each dot proportional to the number of observations for each specific value (count key below each panel). The panels on the right show the distribution of

100 bootstrap sample rank means, where the blue areas under the curve illustrate the 0.025 and 0.975 percentiles of the distribution. A difference is considered significant if the ranked mean for a given condition falls within or beyond the blue percentile of the mean distribution of the wild-type control.
(TIF)

**S3 Fig. In contrast to the widely conserved NRC3, NRC1 is not present in *N. benthamiana*.** Phylogenetic tree of the NRC-helper NLR family based on the full-length protein sequences was inferred using an approximately Maximum Likelihood method as implemented in FastTree [68] based on the Jones-Taylor-Thornton (JTT) model [71]. The tree was rooted on XP_00428175. The different NRC subfamilies are indicated. Domain architecture was determined using NLRtracker [3]. Probable pseudogenes or partial genes, including *Nicotiana* NRC1 are indicated.
(TIF)

**S4 Fig. Statistical analysis of NRC complementation of Cf-4/Avr4-triggered cell death *N. benthamiana nrc2/3/4* CRISPR lines. A**) Statistical analysis was conducted using besthr R package [72]. The dots represent the ranked data and their corresponding means (dashed lines), with the size of each dot proportional to the number of observations for each specific value (count key below each panel). The panels on the right show the distribution of 100 bootstrap sample rank means, where the blue areas under the curve illustrate the 0.025 and 0.975 percentiles of the distribution. A difference is considered significant if the ranked mean for a given condition falls within or beyond the blue percentile of the mean distribution of the wild-type control. **B**) Cf-4/Avr4-triggered hypersensitive cell death is complemented in the *nrc2/3/4* CRISPR lines with NRC3 and to a lesser extent NRC2, but not NRC1 or NRC4. Representative *N. benthamiana* leaves infiltrated with appropriate constructs were photographed 7–10 days after infiltration. The receptor/effector pair tested, Cf-4/Avr4 and Prf (Pto/AvrPto), are labelled above the leaf of NRC CRISPR line *nrc2/3/4*-210.5.5.1. The NRC used for complementation or EV control are labelled on the leaf image.
(TIF)

**S5 Fig. Tomato and pepper NRC3 complement Cf-4/Avr4-triggered hypersensitive cell death in the *nrc2/3/4* knock-out lines. A**) Cf-4/Avr4-triggered hypersensitive cell death is complemented in the *nrc2/3/4* lines with tomato and pepper NRC3 (*Sl*NRC3 and *Ca*NRC3), but not tomato or pepper NRC1 (*Sl*NRC1 and *Ca*NRC1), or NRC2 (*Sl*NRC2 and *Ca*NRC2). Representative *N. benthamiana* leaves infiltrated with appropriate constructs were photographed 7–10 days after agroinfiltration. The receptor/effector pair tested, Cf-4/Avr4 or Cf-4/Avr2 as a negative control, and the NRC used for complementation are labelled on the leaf image. Hypersensitive cell-death is indicated by magenta circles, whereas combinations not displaying hypersensitive cell-death are marked with cyan circles. **B**) *Sl*NRC1, *Sl*NRC2, and *Sl*NRC3 and *Ca*NRC1, *Ca*NRC2, and *Ca*NRC3 protein accumulation. Immunoblot analysis of transiently expressed C-terminally mEGFP-tagged NRCs together with the p19 silencing inhibitor 5 days after agroinfiltration in wild-type *N. benthamiana* plants detected using anti-GFP antibody. P19 alone was used as an empty vector control. **C**) Quantification of hypersensitive cell death. Cell death was scored based on a 0–7 scale between 7–10 days post infiltration. The results are presented as a dot plot, where the size of each dot is proportional to the count of the number of samples with the same score within each biological replicate. The experiment was independently repeated three times. The columns correspond to the different biological replicates. Significant differences between the conditions are indicated with an asterisk (*). Some of the codon-optimized NRC constructs display weak autoactivity. **D**) Statistical analysis

was conducted using besthr R package [72]. The dots represent the ranked data and their corresponding means (dashed lines), with the size of each dot proportional to the number of observations for each specific value (count key below each panel). The panels on the right show the distribution of 100 bootstrap sample rank means, where the blue areas under the curve illustrate the 0.025 and 0.975 percentiles of the distribution. A difference is considered significant if the ranked mean for a given condition falls within or beyond the blue percentile of the mean distribution of the wild-type control.

(TIF)

**S6 Fig. NRC3 complements Cf-2/Rcr3/Avr2-, Cf-5/Avr5-, and Cf-9/Avr9-triggered hypersensitive cell death in the *nrc2/3/4* knock-out lines. A**) Cf-2/Rcr3/Avr2-, Cf-5/Avr5-, and Cf-9/Avr9-triggered hypersensitive cell death is complemented in the *nrc2/3/4* lines with NRC3, while Cf-9/Avr9-triggered hypersensitive cell death is partially complemented with NRC2. Representative *N. benthamiana* leaves infiltrated with appropriate constructs were photographed 7–10 days after agroinfiltration. The receptor/effector pair tested, Cf-2/Rcr3/Avr2, Cf-5/Avr5, and Cf-9/Avr9, are labelled above the leaf of *nrc2/3/4*-210.4.3.1. The NRC used for complementation or EV control are labelled on the leaf image, R3a/Avr3a was used as a positive control. **B**) Quantification of hypersensitive cell death. Cell death was scored based on a 0–7 scale between 7–10 days post infiltration. The results are presented as a dot plot, where the size of each dot is proportional to the count of the number of samples with the same score within each biological replicate. The experiment was independently repeated three times. The columns correspond to the different biological replicates. Significant differences between the conditions are indicated with an asterisk (*). **C**) Statistical analysis was conducted using besthr R package [72]. The dots represent the ranked data and their corresponding means (dashed lines), with the size of each dot proportional to the number of observations for each specific value (count key below each panel). The panels on the right show the distribution of 100 bootstrap sample rank means, where the blue areas under the curve illustrate the 0.025 and 0.975 percentiles of the distribution. A difference is considered significant if the ranked mean for a given condition falls within or beyond the blue percentile of the mean distribution of the wild-type control.

(TIF)

**S7 Fig. Statistical analysis of NRC MADA mutant complementation of Cf-4/Avr4-triggered cell death *N. benthamiana nrc2/3/4* CRISPR lines. A**) Statistical analysis was conducted using besthr R package [72]. The dots represent the ranked data and their corresponding means (dashed lines), with the size of each dot proportional to the number of observations for each specific value (count key below each panel). The panels on the right show the distribution of 100 bootstrap sample rank means, where the blue areas under the curve illustrate the 0.025 and 0.975 percentiles of the distribution. A difference is considered significant if the ranked mean for a given condition falls within or beyond the blue percentile of the mean distribution of the wild-type control. **B**) NRC MADA mutants cannot complement Cf-4/Avr4 or Pto/AvrPto-triggered hypersensitive cell death in the *N. benthamiana nrc2/3/4* CRISPR lines. Representative *N. benthamiana* leaves infiltrated with appropriate constructs were photographed 7–10 days after infiltration. The NRCs tested, NRC2 and NRC3, are labelled above the leaf of NRC CRISPR line *nrc2/3/4*-210.5.5.1. The receptor/effector pair tested, Cf-4/Avr4 and Prf (Pto/AvrPto), are labelled on the leaf image. To ensure the NRC MADA mutants were not autoactive they were expressed with either Cf-4 or Pto and an EV control was taken along.

(TIF)

**S8 Fig. Statistical analysis of NRC ZAR1 α1 helix chimaera complementation of Cf-4/ Avr4-triggered cell death *N. benthamiana nrc2/3/4* CRISPR lines. A**) Statistical analysis was conducted using besthr R package [72]. The dots represent the ranked data and their corresponding means (dashed lines), with the size of each dot proportional to the number of observations for each specific value (count key below each panel). The panels on the right show the distribution of 100 bootstrap sample rank means, where the blue areas under the curve illustrate the 0.025 and 0.975 percentiles of the distribution. A difference is considered significant if the ranked mean for a given condition falls within or beyond the blue percentile of the mean distribution of the wild-type control. **B**) NRC ZAR1 α1 helix chimaeras can complement Cf-4/ Avr4 and Pto/AvrPto-triggered hypersensitive cell death in the *N. benthamiana nrc2/3/4* CRISPR lines. Representative *N. benthamiana* leaves infiltrated with appropriate constructs were photographed 7–10 days after infiltration. The receptor/effector pair tested, Cf-4/Avr4 and Prf (Pto/AvrPto), are labelled above the leaf of NRC CRISPR line *nrc2/3/4-210.5.5.1*. The NRCs tested, NRC2 and NRC3, are labelled on the leaf image. To ensure the NRC ZAR1 α1 helix chimaeras were not autoactive they were expressed with either Cf-4 or Pto and an EV control was taken along.
(TIF)

**S9 Fig. The NRC2 α1 helix can functionally replace the NRC3 α1 helix for Cf-4/Avr4-triggered hypersensitive cell death. A**) Residues 1–17 of NRC2 and 1–21 of NRC3 were replaced with residues 1–21 of NRC3 and 1–17 of NRC2 to generate the NRC2$^{NRC3\alpha1}$ and NRC3$^{NRC2\alpha1}$ chimaeras. NRC3 with the NRC2 α1 helix (NRC3$^{NRC2\alpha1}$) can complement Cf-4/Avr4 and Pto/ AvrPto-triggered hypersensitive cell death in the *N. benthamiana nrc2/3/4* CRISPR lines, while NRC2 with the NRC3 α1 helix (NRC2$^{NRC3\alpha1}$) cannot. Representative *N. benthamiana* leaves infiltrated with appropriate constructs were photographed 7–10 days after infiltration. The receptor tested, Cf-4 and Prf (Pto), is labelled above the leaf of NRC CRISPR line *nrc2/3/4-210.4.3.1*. The NRC and NRC chimaeras tested as well as the effectors are labelled on the leaf image. To ensure the NRC α1 helix chimaeras were not autoactive when expressed with either Cf-4 or Pto and an Avr2 or EV control was taken along, respectively. **B)** Quantification of hypersensitive cell death. Cell death was scored based on a 0–7 scale between 7–10 days post infiltration. The results are presented as a dot plot, where the size of each dot is proportional to the count of the number of samples. Significant differences between the conditions are indicated with an asterisk (*). **C**) Statistical analysis was conducted using besthr R package [72]. The dots represent the ranked data and their corresponding means (dashed lines), with the size of each dot proportional to the number of observations for each specific value (count key below each panel). The panels on the right show the distribution of 100 bootstrap sample rank means, where the blue areas under the curve illustrate the 0.025 and 0.975 percentiles of the distribution. A difference is considered significant if the ranked mean for a given condition falls within or beyond the blue percentile of the mean distribution of the wild-type control.
(TIF)

**S10 Fig. NRC3 E15Q can complement Cf-4/Avr4-triggered hypersensitive cell death. A**) NRC3 and the NRC3 E15Q mutant can complement Cf-4/Avr4 and Pto/AvrPto-triggered hypersensitive cell death in the *N. benthamiana nrc2/3/4* CRISPR lines. Representative *N. benthamiana* leaves infiltrated with appropriate constructs were photographed 7–10 days after infiltration. The receptor/effector pair tested, Cf-4/Avr4 and Prf (Pto/AvrPto), as well as the NRC and NRC mutants used are labelled on the leaf image. To ensure the NRC3 E15Q mutant was not autoactive we expressed it with either Cf-4 or Pto and an EV control. **B)** Quantification of hypersensitive cell death. Cell death was scored based on a 0–7 scale between 7–10 days post infiltration. The results are presented as a dot plot, where the size of each dot is

proportional to the count of the number of samples with the same score. Significant differences between the conditions are indicated with an asterisk (*). **C**) Statistical analysis was conducted using besthr R package [72]. The dots represent the ranked data and their corresponding means (dashed lines), with the size of each dot proportional to the number of observations for each specific value (count key below each panel). The panels on the right show the distribution of 100 bootstrap sample rank means, where the blue areas under the curve illustrate the 0.025 and 0.975 percentiles of the distribution. A difference is considered significant if the ranked mean for a given condition falls within or beyond the blue percentile of the mean distribution of the wild-type control.
(TIF)

**S11 Fig. Statistical analysis of Cf-4/Avr4-suppression mediated by divergent pathogen effectors. A**) Statistical analysis was conducted using besthr R library [72]. The dots represent the ranked data and their corresponding means (dashed lines), with the size of each dot proportional to the number of observations for each specific value (count key below each panel). The panels on the right show the distribution of 100 bootstrap sample rank means, where the blue areas under the curve illustrates the 0.025 and 0.975 percentiles of the distribution. A difference is considered significant if the ranked mean for a given condition falls within or beyond the blue percentile of the mean distribution of the wild-type control. **B**) Cf-4/Avr4-triggered hypersensitive cell death is suppressed by SPRYSEC15 and AVRcap1b, but not the EV control. Representative image of *N. benthamiana nrc2/3/4*-210.5.5.1 CRISPR line leaves which were agroinfiltrated with NRC3/suppressor constructs, as indicated above the leaf, and either autoactive NRC3$^{D480V}$, Prf (Pto/AvrPto), R3a/Avr3a, or Cf-4/Avr4 as labelled on the leaf image, photographed 7–10 days after infiltration.
(TIF)

**S1 Table. Description of constructs and *Agrobacterium* strains.**
(XLSX)

**S1 Dataset. MAFFT alignment of the unique full-length NCBI RefSeq NRCs (Fasta format).** This file contains the MAFFT alignment of 134 unique NRCs found in the NCBI RefSeq database. XP_00428175 was added as an outgroup.
(FASTA)

**S2 Dataset. Approximately maximum likelihood phylogenetic tree of the NCBI RefSeq NRCs (Nexus format).** This file contains the phylogenetic analysis of the full-length NRCs found in the NCBI RefSeq database using the JTT method.
(NEX)

## Acknowledgments

We thank Aleksandra Białas, Ana Cristina Barragan Lopez, Clémence Marchal, Paul Crosnier, and Thorsten Langner for their helpful comments on the figures, and Phil Robinson for photography.

## Author Contributions

**Conceptualization:** Jiorgos Kourelis, Daniel Lüdke, Hiroaki Adachi, Chih-Hang Wu, Sophien Kamoun.

**Data curation:** Jiorgos Kourelis, Mauricio P. Contreras, Chih-Hang Wu.

**Formal analysis:** Jiorgos Kourelis.

**Funding acquisition:** Sophien Kamoun.

**Investigation:** Jiorgos Kourelis, Mauricio P. Contreras, Adeline Harant, Hsuan Pai, Daniel Lüdke, Chih-Hang Wu.

**Project administration:** Sophien Kamoun.

**Visualization:** Jiorgos Kourelis, Hsuan Pai.

**Writing – original draft:** Jiorgos Kourelis, Sophien Kamoun.

**Writing – review & editing:** Jiorgos Kourelis, Mauricio P. Contreras, Adeline Harant, Daniel Lüdke, Hiroaki Adachi, Lida Derevnina, Chih-Hang Wu, Sophien Kamoun.

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
