## [Decision Letter · Decision Letter 0]

24 Mar 2022

Dear Dr Kamoun,

Thank you very much for submitting your Research Article entitled 'The helper NLR immune protein NRC3 mediates the hypersensitive cell death caused by the cell-surface receptor Cf-4' to PLOS Genetics.

The manuscript was fully evaluated at the editorial level and by independent peer reviewers. The reviewers appreciated the attention to an important problem, but raised some substantial concerns about the current manuscript. Based on the reviews, we will not be able to accept this version of the manuscript, but we would be willing to review a much-revised version. We cannot, of course, promise publication at that time.

If you decide to revise the manuscript for further consideration at PLOS Genetics, please aim to resubmit within the next 60 days, unless it will take extra time to address the concerns of the reviewers, in which case we would appreciate an expected resubmission date by email to plosgenetics@plos.org.

[LINK]

We are sorry that we cannot be more positive about your manuscript at this stage. Please do not hesitate to contact us if you have any concerns or questions.

Yours sincerely,

Tiancong Qi

Associate Editor

PLOS Genetics

Claudia Köhler

Section Editor: Plant Genetics

PLOS Genetics

Comments from the Associate Editor:

Generally, this work provides valuable data supporting NRC3 function in Cf4/Avr4-induced cell death, which will be of great interest for the field. However, I agree with several of the reviewers' comments that should be addressed to improve the manuscript. Specifically, I consider the following points necessary for improving the present manuscript: 1) As reviewer 3 suggests, analyzing variation pattern of helper NRCs, studying the α1 helix feature and MADA motif function will help to understand NRCs protein function. It would furthermore be important to determine NRC3's specificity in Cf4/Avr4-induced cell death. The data show that the ZAR1 α1 helix substitution for the NRC2 N-terminal equivalent sequence allows NRC2 to activate Cf4/Avr4-triggered cell death. It would be important to address whether the α1 helix of NRC3 determines its functional specificity. Furthermore, it would be relevant to test whether NRC3's α1 helix substitution can also allow NRC2 to activate Cf4/Avr4-triggered cell death. 2) More evidences are needed, as reviewer 1 suggests, to show that NRCs may function similarly with ZAR1 as cation channel and form resistosomes. It would strengthen the manuscript if additional experimental evidence (e.g. testing whether point mutations of conserved amino acids within NRC3 affect its function, whether NRC3 oligomerizes after Cf4/Avr4 activation, whether NRC3 relocates to the plasma membrane after Cf4/Avr4 activation) would be provided to illustrate the molecular function of NRC3 on mediating Cf4/Avr4-induced cell death. 

Reviewer's Responses to Questions

**Comments to the Authors:**

Reviewer #1: This study is examining the involvement of NRC2/3 and 4 in cf-4 signaling. It was already known that NRCs are involved in Cf-4 cell death but the current study brings more detailed information on the specific contribution of NRC3 and to a lesser extent NRC2 to Cf-4 signaling. However, only cell death is evaluated, using a semi-quantitative scale, and in an overexpression system. This analysis is limited and could be complemented with a quantitative analysis of cell death and defense gene expression. The authors then show that NRC3 cell death is suppressed by mutations affecting some conserved residues in N-term and that ZAR1 N-term could rescue the phenotype, suggesting that NRCs and ZAR1 function similarly. This data does not show that NRC3 could be a cation channel. However, simple experiments could greatly enhance the novelty and quality of the study. ZAR1 E11, which is specifically involved in calcium permeation, is conserved in NRC3. Does mutating this residue to a Q in NRC3 affect Cf-4 cell death (or NRC3auto-activity)? Is NRC3 cell death inhibited by calcium channel blockers? Does auto active NRC3 trigger calcium influx and if yes, does it requires E14? If those experiment were conducted, they would link NRC3 to the regulation of calcium level which would be a considerable advance for the understanding of NRC3 function. Importantly, the text needs extensive editing to increase clarity and precision, in particular in the introduction and discussion.

If those changes were made, this study would be very valuable to the field.

Major comments:

Abbreviated terms must be spelled at first use. I noticed ZAR1, RPM1, ADR1, NRG1, MAPK etc

The introduction should be made more accessible to non-specialist, for instance:

“In Arabidopsis, the EDS1/PAD4/ADR1 and, to a lesser extent, EDS1/SAG101/NRG1 modules are genetically required for a subset of the immune responses triggered by LRR-RPs” and latter “Since the EDS1/PAD4/ADR1 and EDS1/SAG101/NRG1 modules are typically associated with hypersensitive cell death signaling, this raises the question of what the molecular mechanism of this interaction is.” These proteins have not been introduced and defined before. Why would we expect these modules to be involved in cf-4 cell death? Is it canonical for PRR to trigger cell death? A non-specialist reader cannot understand.

The introduction also should be more focused and clearly present the background knowledge required to understand the motivation and results of the study. I would appreciate a reminder of which NRC is required for which sensor NLR function. Extensive editing is required.

Line 76 “specific variant cyclic ADP-ribose (v-cADPR)” The study of Yu D. et al 2021 (TIR domains of plant immune receptors are 2′,3′-cAMP/cGMP synthetases mediating cell death) should be mentioned.

Line 169 “NRC3 is conserved in the Solanaceae and likely mediates hypersensitive cell death triggered by a variety of leucine-rich repeat receptor-like proteins. We propose that NRC3 is a core node that connects cell surface and intracellular immune networks” What other receptors did you study here? What, in this study, justify such a broad proposition?

Figure 4: precise the residues used for the constructs in the figure, the caption and the corresponding results section. The author should be very cautious in the interpretation of the data as nothing tells us that the cell death triggered by the WT NRC3 and the chimeric NRC3 are the result of the same molecular mechanism. ZAR1 alpha1 helix is not sufficient to form a cation channel, the chimeric NRC3 data only indicates that NRC3 is still functional, even with ZAR1 alpha1 helix. Supporting data showing NRC3 regulates (directly or not) calcium levels in the cytoplasm and data showing that calcium influx is required for NRC3 function would be valuable. In addition, some residues in ZAR1 and in NRG1 and ADR1 have been shown to be specifically involved in calcium influx. Studying the orthologous residue in NRC3 would be a better indication that NRC3 forms a resistosome similar to ZAR1.

Minor comments

Introduction. Please avoid the use of unspecific terms like “generally" and "typically", be more precise. Also, avoid stating what "the community" thinks, please stick to the facts and let me make my own mind. I noticed (among others):

Line 38 “the parasite is blocked” precise

“The ligand sensing mechanisms are generally well-known” on line 52. This sentence does not bring information (generally) and is highly debatable (well-known).

Line 55 and 56 “both classes of receptors induce common signaling pathways” imprecise or incorrect

Line 60 “intracellular NLRs differs and converges is one of the main open questions in the field of plant immunity”

Line 68 “Much progress has been made in recent years in understanding the molecular mechanisms of recognition and signaling of plant NLRs”

Line 69 “In general,”

Line 144 “NRCs form redundant nodes in complex receptor networks, with different NRCs exhibiting different specificities for their sensor NLRs.” So, NRCs are redundant and specific?

Line 160 “Using a combination of reverse genetics and complementation, we determined that Cf-4/Avr4-triggered cell death is largely mediated by NRC3, and not NRC1.” NRC1? Which organism?

Line 189 “indicating that the reduction in cell death is not correlated with Cf-4” imprecise.

Line 190 “Cf-4/Avr4 expression resulted in a specific chlorosis in the nrc2/3 and nrc2/3/4 lines which did not develop into confluent cell death (Figure 1C, Figure S1B).” What is specific about this chlorosis? Please go straight to the point.

Line 244 “only weakly complemented Cf-4/Avr4-triggered cell death” Avoid the use of unspecific terms.

Line 300 “NRC2L17E and NRC3L21E correspond to the NRC4L17E mutant” Imprecise

Line 352 “In the activated ZAR1 resistosome, the N-terminal α1 helix of the CC domain forms a Ca2+-permeable membrane-permeable pore” delete the extra permeable.

Line 511 “Additionally, the ZAR1 α1 helix can functionally substitute for the N-terminal equivalent sequence of NRC3” This is said two sentence earlier

Line 521 “It may be that this chimeric protein is more “trigger happy”, either due to reduced autoinhibition or enhanced capacity to generate the resistosome funnel.” We can only say the chimeric construct is a gain of function. We don’t know if the effect is coming from increased activation (trigger happy) or more efficient activation of downstream signaling component.

Reviewer #2: The author's group performed a series of work on the identification and characterization of helper NLRs in the NLR network. They discovered several helper NLRs including NRC2, NRC3 and NRC4 in Solanaceae plants that are required for the sensor NLR-mediated immune signaling (Wu et al., 2017, PNAS); They further screened a large amounts of effectors from diverse pathogens that can target and suppress the activity of NRCs, and revealed plant pathogens evolves to counteract the NRCs hub to subvert the NLR-mediated defense (Derevnina et al., 2021, PLoS Biology). This work further extends these studies by investigating the functional role of NRCs in pattern-triggered immunity (PTI) besides the intracellular effector-trigged immunity (ETI) as they reported previously. They establish a functional link between the cell death caused by PTI and the NLR resistosome. Overall, this work is interesting and further deepen our understanding of intricate immune receptor network in plants.

Major concerns

1. What is the biological significance of the convergent NRC hub? On one hand, it is proposed that it would be beneficial for the plant to defend against diverse pathogens by using one system. However, just like the double-edge, studies from the author's group indicated that the convergent NRC hub could be easily be targeted by the pathogens, leading to the disruption of the immune barrier. This is an arm race between plant and pathogens. Figure 5B，the author used some suppressors (AVRcap1b and SPRYSEC15) to further verify the specific role of NRC3 in Cf4/Avr4-induced cell death, the design of this experiments is good. I wonder whether these events could occur under natural conditions? Have the author observed some mixed infections of some pathogens in plants under natural conditions, and the plants displayed an increased symptom severity compared with infection of a pathogen alone? and does it associate with the suppression of the NRC hub?

2. How does or what is mechanism underlying these effectors-mediated suppression of NRC activity. In fact, there remains largely unknown whether AVRcap1b and SPRYSEC15 inhibit the the formation of resistosome or the subsequent immune signal transduction, the author may discuss this.

3. What is the relationship between Cf-4 and NRC3? Does Cf-4 interacts with NRC3?

Reviewer #3: Kourelis and colleagues present the work investigating the role of helper NLR, NRCs in N. benthamiana, in mediating HR triggered by the cell-surface receptor Cf-4. Genetic components required for Cf-4/Avr4-mediated HR had been analyzed in N. benthamiana using VIGS (Gabriëls, Takken et al., 2006, MPMI), identifying an NLR, which was named as NRC1 back then. Given that there are recent advances in the field on uncovering the role of helper NLRs (such as ADR1-EDS1-PAD4 by Pruitt et al.) serving as a converging point interconnecting signaling from surface-bound receptors and intracellular NLR receptors, the presented work on Cf-4 requiring one of the helper NLRs previously characterized in Solanaceae is quite timely to corroborate the helper’s role in immune signaling. This work utilized the previously generated stable transgenic lines of N. benthamiana carrying mutations in multiple NRC genes and wisely employed the transient complementation system to reintroduce individual NRC genes as wild-type or mutated versions to address the functional specificity. The authors successfully identified NRC3 as the main contributor of the Cf-4-mediated HR, while its homolog NRC2, despite no complementation as a wild-type construct, could be expanded to contribute to the Cf-4-mediated HR signaling when the alpha1 helix was replaced with the one from Zar1. The authors also employed known effectors that suppress the helper NRCs to negate the function of NRC3 participating to Cf-4-mediated HR. These findings collectively suggest that the helper NLR function and its specificity in the participating signaling pathways can be modified for future genetic engineering. All these conclusions were made with quantitation of the transient HR assays in N. benthamiana along with the confirmation of protein expression at adequate level.

This work is timely addition to the field in characterizing the signaling potential of helper NLRs.

Generally, I do not find major flaw or amendment points, however, I wish I could find robust discussion on potential mechanistic explanation on this convergence. Especially, NRC3, which is claimed to be conserved, shall be discussed in terms of the variation pattern as compared with other helper NRCs. As the alpha 1 helix feature appears to matter, and MADA motif is rather loose and remains to be further defined, this work may have a saying in refining the functional MADA motif. In this sense, I would propose to look into sequence alignments of NRCs (or provide previous works related to this) in various species to further define functional motifs in NRCs, such as pinpointing the variation pattern in the MADA motif.

In addition, I propose to include exact positions of alpha1-swapping with numbered AA positions in Figure 4 A and B. I do not find that information in the main text or in main method section. For example, the statement in L355-356 and Figure 4A do not match; As Figure 4A does not show the sequence alignment, but structure, it would be impossible to see the homology in the presented figure.

In the following section, I provide comments that the authors could use to tighten the loose ends in their interpretation of datasets or fix typographic errors.

L140: R genes encoding (instead of encoded) sensor NLRs

L149: add space after [42]

L157: required for downstream signaling upon LRR-RP activation

L162: As NRC1 from tomato was introduced in L133 as a Cf-4 signaling factor, it would be relevant to specify the NRCs (NRC3 and NRC1) are from N. benthamiana.

L178: It would be relevant to state clearly here that the lines employed in this study were generated previously. The current sentence reads as if these were generated for this work despite the references.

Listing specific lines available (or used) would also help, instead of stating it as “various” combinations. Only three different lines were used in this work, which should be relevant to introduce one by one.

L239: italicizes SI in SINRC1 if the authors want to be consistent with L245.

Figure 3A: The alignment of NRCs with ZAR1 seems not well spaced, for example, I in NRC3 is not positioned in the middle of the space and the VV residues in Zar1 is squeezed so that the second V is cut off at the end. Using the font courier may solve this problem.

L291-311: experiments were carried out both with NRC2 and NRC3 to test the function of MADA but the conclusion was only made for NRC3 in L310-311. Please clarify the use of NRC2L17E constructs in line of NRC3 results. I also wondered where the previous notion (L307) is to be found. It would be necessary to add a pointer to indicate actual dataset presented in a previous figure (Figure 2 and 3S?).

L351-352: “permeable” is duplicated. Would it be the best phrasing for describing the Zar1 action?

L369-370: This conclusion should be more specific than the current simple version. The chimeric constructs do not merely “replace” but apparently augmented the function in NRC2 in particular. The authors have a tendency of making conclusions of NRC2 and NRC3 together, while there are distinct spectrum in helper function. This tendency of generalizing should be fixed.

L419: Be consistent with citation format (numbers in bracket is commonly found).

L494: The same comment is given to the authors as L162. Please specify species name for the previous work on NRC1. If the previous naming is confusing (indeed it is referring to an enigmatic entity due to VIGS), the nomenclature is to be corrected where relevant and the usage of NRC1 should be avoided thereafter.

**Have all data underlying the figures and results presented in the manuscript been provided?**

Reviewer #1: Yes

Reviewer #2: Yes

Reviewer #3: Yes

PLOS authors have the option to publish the peer review history of their article (what does this mean?). If published, this will include your full peer review and any attached files.

Reviewer #1: No

Reviewer #2: No

Reviewer #3: **Yes: **Eunyoung Chae

---

## [Decision Letter · Decision Letter 1]

27 Jul 2022

Dear Dr Kamoun,

Thank you very much for submitting your Research Article entitled 'The helper NLR immune protein NRC3 mediates the hypersensitive cell death caused by the cell-surface receptor Cf-4' to PLOS Genetics.

The manuscript was fully evaluated at the editorial level and by independent peer reviewers. The reviewers appreciated the attention to an important topic but identified some concerns that we ask you address in a revised manuscript. Specifically, we agree with reviewer 1 that the present data are insufficient to suggest that NRC3 forms a resistosome and functions as a calcium channel like ZAR1. In a revised manuscript we therefore advice to tone down the related statements unless more experimental data is provided. The related descriptions like “resistosome” and “Ca2+ channel” concerning NRC3 molecular function should be weakened and revised including the model (Fig 6).

We therefore ask you to modify the manuscript according to the review recommendations. Your revisions should address the specific points made by each reviewer.

[LINK]

Yours sincerely,

Tiancong Qi

Academic Editor

PLOS Genetics

Claudia Köhler

Section Editor

PLOS Genetics

The revised manuscript has largely improved and addresses most issues raised by reviewers. However, I agree with reviewer 1 that the present data are insufficient to suggest that NRC3 forms a resistosome and functions as a calcium channel like ZAR1. In a revised manuscript the authors should therefore tone down the related statements unless more experimental data is provided. The related descriptions like “resistosome” and “Ca2+ channel” concerning NRC3 molecular function should be weakened and revised including the model (Fig 6).

Reviewer's Responses to Questions

**Comments to the Authors:**

Reviewer #1: This study characterizes the involvement of NRCs in PRR-triggered cell death. As is, it is an interesting study presenting overall valuable data. Addition of the pepper and tomato NRC data is interesting and does extend the genetic characterization of the NRC family. The authors chose to only consider cell death as a proxi for NRC function and do not wish to investigate gene expression regulation. It would be interesting but I think it is fine since the conclusion related to NRC function are explicitly limited to cell death.

I think this manuscript could be published without additional experiments if the authors were not speculating on the molecular function of NRC3, resistosome formation and calcium influx, from the data presented in figure 4. The experimental approach used in figure 4 is limited and the data is overinterpreted.

I understand biochemical analyses of NLRs are technically challenging but if they are not conducted, the term resistosome should not be used to describe active NRC3 complexes. NLRs act through various mechanisms: holoenzyme formation, ion channel formation or by recruiting downstream signaling components. It is early to state that CNLs all act by forming ZAR1-like resistosomes from just two examples. The chimera experiment does not give us information on the WT protein function in the absence of data on oligomerization, membrane localization and calcium influx (ZAR1 alpha 1 helix is also required for plasma membrane localization so the ZAR1 helix could complement loss of proper subcellular localization of NRC3 mada mutant). Contrary to what is stated in the text (line 336), the alpha 1 helix does not form a pore, not alone. It only contributes to 1/3 of the ZAR1 channel. So, the chimeric constructs do not prove that NRC3 oligomerize in a resistosome-like oligomer. The fact that E14Q NRC3 mutant is not a loss of function (in these assays using overexpression and only considering cell death) does not suggest that NRC3 functions like ZAR1. Several other negatively charged residues are present in the alpha1 helix and could still be required for function. These could be tested alone or in combination. If NRC3 activity was correlated to calcium influx and if NRC3 ZAR1alpha1 E11Q mutant lost cell death, it would indicate that the function complemented by the alpha 1 helix is calcium influx. Then, I would cautiously admit that NRC3 likely forms a resistosome such as ZAR1. These experiments are not technically challenging.

I noted 2 typos:

Line 259 : Cf-4/Avr4 cell death (Fig 2C and 2D, S4A Fig) for statistical analysis

Line 345: Like wild type NRC3,

Reviewer #2: The author has addressed my concern.

Reviewer #3: The authors had successfully addressed major issues raised in the first round of evaluation. I was glad to see the issues on NRC redundancy and specificity has been corroborated with new data. Writing and organization have much improved so that the revised manuscript stands as a nice report of the new finding.

The work on NRCs described in this manuscript is quite timely and an important addition to the field in terms of broadening the role NRC helper function in immune signaling upon RLP activation. The presented work will attract broad audience who is well versed in genetics of immunity, disease resistance, immune signaling, who would always like to read PLoS Genetics articles for inspirational genetics studies. I would agree with other reviewers and the editor that there are some room for improvement in regards to the mechanistic action of NRCs, however, I do not think that detailed mechanistic action is necessary to be included to corroborate genetics experiments. Many times, nice genetic experiments tell much more than we expected to position a factor under a singling pathway, and this piece of work delivers all the necessary points.

I only provide comments that may help clarifying some textual ambiguity.

L165: to me, partial phenotypes are rather more vague than weak phenotypes. Would it mean “partial suppression” or “weak suppression” of Cf-4 mediated cell death upon silencing of NRCs?

L191-192: in L163, the author already mentioned that there is “no” true NRC1 ortholog in N. benthamiana. The way it is descried such as it is NRC3 but “not a N. benthamiana NRC1 homolog as previously implied” is confusing. The argument on "A but not B" applies when A and B are separate entities. Here, the authors rather nicely clarified that previously implied A is A’. Again, here this work identifies that NRC3 in N. benthamiana is the previously suggested NRC1 in N. benthaminan, sorting out confusion by refining the function. I would suggest the authors to make it clear here by revising the sentence (after comma), such as

“NRC3 homolog, clearly defining loosely identified a cell death controlling NLR functioning downstream of the Cf-4 RLP”.

**Have all data underlying the figures and results presented in the manuscript been provided?**

Reviewer #1: Yes

Reviewer #2: None

Reviewer #3: Yes

PLOS authors have the option to publish the peer review history of their article (what does this mean?). If published, this will include your full peer review and any attached files.

Reviewer #1: No

Reviewer #2: No

Reviewer #3: **Yes: **Eunyoung Chae

---

## [Editor Report · Decision Letter 2]

6 Sep 2022

Dear Dr Kamoun,

We are pleased to inform you that your manuscript entitled "The helper NLR immune protein NRC3 mediates the hypersensitive cell death caused by the cell-surface receptor Cf-4" has been editorially accepted for publication in PLOS Genetics. Congratulations!

Yours sincerely,

Tiancong Qi

Academic Editor

PLOS Genetics

Claudia Köhler

Section Editor

PLOS Genetics

Comments from the reviewers (if applicable):

**Data Deposition**

http://datadryad.org/submit?journalID=pgenetics&manu=PGENETICS-D-22-00212R2

**Press Queries**

---

## [Editor Report · Acceptance letter]

16 Sep 2022

PGENETICS-D-22-00212R2 

The helper NLR immune protein NRC3 mediates the hypersensitive cell death caused by the cell-surface receptor Cf-4 

Dear Dr Kamoun, 

We are pleased to inform you that your manuscript entitled "The helper NLR immune protein NRC3 mediates the hypersensitive cell death caused by the cell-surface receptor Cf-4" has been formally accepted for publication in PLOS Genetics! Your manuscript is now with our production department and you will be notified of the publication date in due course.

With kind regards,

Zsofia Freund

PLOS Genetics

On behalf of:
